# DrVoice: Parallel Speech-Text Voice Conversation Model via Dual-Resolution Speech Representations

**Chao-Hong Tan, Qian Chen, Wen Wang, Chong Deng, Qinglin Zhang,
Luyao Cheng**, **Hai Yu**, **Xin Zhang**, **Xiang Lv**, **Tianyu Zhao**, **Chong Zhang**,
**Yukun Ma**, **Yafeng Chen**, **Hui Wang**, **Jiaqing Liu**, **Xiangang Li**, **Jieping Ye**
Tongyi Fun Team, Alibaba Group
{tanchaohong.ch, tanqing.cq, w.wang}@alibaba-inc.com

## Abstract

Recent studies on end-to-end (E2E) speech generation with large language models (LLMs) have attracted significant community attention, with multiple works extending text-based LLMs to generate discrete speech tokens. Existing E2E approaches primarily fall into two categories: (1) Methods that generate discrete speech tokens *independently* without incorporating them into the LLM's autoregressive process, resulting in text generation being unaware of concurrent speech synthesis. (2) Models that generate *interleaved* or *parallel* speech-text tokens through *joint autoregressive modeling*, enabling mutual modality awareness during generation. This paper presents **DrVoice**, a parallel speech-text voice conversation model based on joint autoregressive modeling, featuring **dual-resolution speech representations**. Notably, while current methods utilize mainly 12.5Hz input audio representation, our proposed dual-resolution mechanism reduces the input frequency for the LLM to **5Hz**, significantly reducing computational cost and alleviating the frequency discrepancy between speech and text tokens and in turn better exploiting LLMs' capabilities. Experimental results demonstrate that DrVoice-7B establishes new state-of-the-art (SOTA) on prominent speech benchmarks including OpenAudioBench, VoiceBench, UltraEval-Audio and Big Bench Audio, making it a leading open-source speech foundation model in ∼7B models. [1]

## 1 Introduction

Developments in spoken dialogue systems are critical to human-computer interaction, as natural human communication inherently relies on verbal exchanges. Recently, Large Language Model (LLM) based spoken dialogue systems, exemplified by systems like GPT-4o (OpenAI, 2024b), demonstrate great potential for seamless and natural interactions with users. LLM-based spoken dialogue systems can be generally categorized into **cascaded systems** and **end-to-end (E2E) systems**, with the distinction lying in whether the backbone LLM can directly comprehend speech representations and generate speech outputs. Early cascaded systems integrate separately trained Automatic Speech Recognition (ASR), LLM, and Text-to-Speech (TTS) modules. Such systems inherently suffer from error accumulation, loss of acoustic details (e.g., emotion), and significant latency. Alternatively, the ASR module could be eliminated by introducing audio understanding foundation models (Chu et al., 2023; 2024). E2E systems have emerged to further eliminate ASR and TTS modules, and establish direct connections between speech representations and LLMs. However, training LLMs to generate highly intelligent speech outputs remains challenging. E2E systems struggle with the quality of speech token generation, primarily due to inefficient utilization of textual information during speech token generation.

---

[1] All source code and the model checkpoint based on enhanced base model are available at `https://github.com/FunAudioLLM/Fun-Audio-Chat`.

Recent works on E2E models to address these challenges have focused on two primary directions (Chen et al., 2024a): **Text-Driven Speech Models** (Yao et al., 2024; Li et al., 2025) and **Joint Speech-Text Models**. Text-Driven Speech Models refer to systems in which LLMs process speech representations as input, produce textual responses, and utilize the LLM's representations as input to a speech decoder for speech generation. In contrast, Joint Speech-Text Models involve LLMs taking speech representations as input and generating *both text tokens and speech tokens simultaneously*. The key distinction lies in whether speech token generation can influence the subsequent generation of text tokens within the LLM: this feedback loop of speech tokens to LLM is present in Joint models but absent in Text-Driven models. Joint Speech-Text Models can be further categorized into **interleaved speech-text modeling** (Zeng et al., 2025; Zhang et al., 2024a) and **parallel speech-text modeling** (Défossez et al., 2024; Chen et al., 2024a; KimiTeam et al., 2025). Interleaved speech-text modeling alternates speech and text representations as inputs to the LLM, while parallel speech-text modeling fuses speech and text representations before feeding them into the LLM.

While both Text-Driven Speech Models and Joint Speech-Text Models explicitly incorporate textual guidance into speech token generation to leverage the LLM's capabilities, they have distinct limitations. The architecture of Text-Driven Speech Models creates a unidirectional information flow where text generation is completed before speech synthesis begins. This prevents the LLM from conditioning its textual output on the generated speech tokens, thus limiting its ability to explore fine-grained paralinguistic attributes like emotion and prosody. On the other hand, Joint Speech-Text Models degrade the original text generation capabilities due to speech token interference, making the preservation of text capabilities a critical challenge. Nevertheless, Joint Speech-Text Models enforce multimodal interaction and generation and empower greater potentials (Chen et al., 2024a); hence, in this work, we focus on enhancing joint speech-text modeling. Recently, Kimi-Audio (KimiTeam et al., 2025) sets a new state-of-the-art (SOTA) in joint speech-text modeling. However, this approach still suffers from notable limitations. It not only requires extensive training data but also incurs significant computational costs due to its 12.5Hz audio representation. Furthermore, the high token rate creates a frequency mismatch with the much lower rate of text tokens ($\sim$3Hz) (Chen et al., 2024a), which can dilute semantic information and consequently hinder the full utilization of the LLM's core capabilities.

In this work, we propose DRVOICE, a novel parallel speech-text voice conversation model with **Dual-Resolution Speech Representations (DRSR)**. For speech comprehension, we introduce a grouping mechanism that maps 25Hz discrete audio tokens to 5Hz speech representations, effectively alleviating the temporal resolution discrepancy between speech and text tokens. During generation, combined speech-text embeddings form the assistant's autoregressive input. The hidden states captured from shared LLM layer are then passed in parallel to a Text Head for text token prediction and a Speech Refined Head (SRH) to generate the corresponding *ungrouped* speech tokens.

To further enhance the model's reasoning and coherence of its output, we incorporate a Chain-of-Modality (CoM) (Zhang et al., 2023a) strategy. CoM prompts the model to first generate a complete response in text, allowing it to structure its thoughts before engaging in parallel speech-text generation. This intermediate reasoning step improves modality alignment and reduces errors. We design system prompts to control the output modes, enabling text-only output, direct parallel speech-text output, or the CoM-enhanced parallel output. We also develop a Core-Cocktail training strategy to refine model optimization and LLM knowledge retention .

Our contributions are three-fold: 1) We propose **DRVOICE**, a novel parallel speech-text conversation model featuring **Dual-Resolution Speech Representations (DRSR)**. This core architectural innovation effectively alleviates the temporal resolution mismatch between speech and text tokens, reducing computational costs (empirically achieving a reduction of nearly 50% in GPU hours for training) and better preserving the LLM's semantic processing capabilities. 2) We introduce two new training strategy, including a **CoM-Mixing training strategy** acting as a curriculum, using the structured CoM reasoning to scaffold speech generation, and a **Core-Cocktail training strategy** for retaining the knowledge of LLMs. 3) Our comprehensive experimental results reveal that DRVOICE-7B achieves new SOTA performance on prominent benchmarks including OpenAudioBench (audio understanding), VoiceBench (benchmarking LLM-based voice assistants), UltraEval-Audio (both speech understanding and generation) and Big Bench Audio (reasoning and understanding capabilities of audio-processing models), solidifying its position as a premier open-source speech foundation model among $\sim$7B models.

## 2 RELATED WORK

**Speech Tokenization.** Two primary directions exist for converting continuous speech signal into processable sequences: continuous representations, e.g., Whisper (Radford et al., 2022), and discrete representations. Since LLMs generate discrete tokens, continuous representations face challenges in direct integration with LLMs for speech generation. In contrast, discrete tokens enable LLMs to handle speech similar to text tokens, and are typically categorized into two categories: 1) acoustic tokens optimized for reconstructing high-quality audio through neural audio codecs (Défossez et al., 2023), and 2) semantic tokens typically derived from *self-supervised* pre-trained models with masked language modeling as training objective (Hassid et al., 2023) or supervised learning as S3Tokenizer using in CosyVoice (Du et al., 2024a;b; 2025), and prioritized for linguistic content (Hsu et al., 2021). While acoustic tokens achieve superior acoustic fidelity in sound reconstruction, semantic tokens demonstrate stronger alignment with semantic representations (Zhang et al., 2023b), thereby facilitating more effective extension of MLLMs' capabilities in both speech comprehension and generation (Zeng et al., 2025; Zhang et al., 2024b; 2023a; Borsos et al., 2023). In this work, we use S3Tokenizer as the speech tokenizer due to its solid semantic capabilities and compatibility with the strong Speech Detokenizer from CosyVoice for synthesis. S3Tokenizer is a supervised semantic speech tokenizer based on the pre-trained SenseVoice-Large model (An et al., 2024) that enhances the semantic relationship of extracted tokens. S3Tokenizer is robust to data noise, and reduces the reliance on clean data.

**E2E Speech Foundation Models.** Text-Driven Speech Models (Fang et al., 2024; Wang et al., 2024; Fu et al., 2025; Yao et al., 2024; Huang et al., 2025; Chen et al., 2025) integrate speech encoder, adapter, LLM, and a streaming speech decoder, and can simultaneously generate text and speech. Qwen2.5-Omni (Xu et al., 2025) employs the Thinker-Talker architecture, with Thinker handling multimodal understanding and text generation and Talker handling speech token production. Since Thinker cannot receive speech tokens during generation, the framework inherently limits awareness of speech token generation states, constraining applications such as full-duplex voice conversation.

Joint Speech-Text Models explore two different architectures, *Interleaved* and *Parallel*. Interleaved models (Zhang et al., 2024a; Zeng et al., 2025; Li et al., 2025; Long et al., 2025; Wu et al., 2025) adopt interleaved decoding to support simultaneous generation of speech and text tokens, while parallel models (Défossez et al., 2024; Xie & Wu, 2024a;b; Chen et al., 2024a; KimiTeam et al., 2025) conduct parallel decoding. The parallel model Moshi (Défossez et al., 2024) employs a compact Deep Transformer to predict $k$ tokens while relying solely on current-step representations. To mitigate limitations of discrete tokens in audio understanding, Kimi-Audio (KimiTeam et al., 2025) introduces dual-tokenizer combining discrete semantic tokens with continuous Whisper (Radford et al., 2023) features, preserving both semantic and acoustic information. In order to further enhance the efficiency of both training and inference and alleviate the issue of granularity misalignment between the text and speech modalities, our approach leverages DRSR to reduce the LLM input frame rate to 5Hz, without sacrificing performance.

## 3 METHODOLOGY

Figure 1 illustrates the architecture of DRVOICE. The system consists of three main components: (1) Speech Encoder and Speech Tokenizer process the speech wave into hidden representations for the user end and the assistant end respectively, (2) a Multimodal Large Language Model (MLLM) consists of shared LLM layer, a Text Head, and a Speech Refined Head (SRH) for token generation, and (3) Speech Detokenizer to generate wave from speech tokens. The system operates through multimodal input encoding and coordinated speech-text output generation. During inference, user inputs (text or speech) are first mapped to a unified semantic space, processed by MLLM to produce parallel speech-text responses through SRH and text head. To effectively train this system, we introduce the CoM-Mixing Training and Core-Cocktail Training strategies.

### 3.1 SPEECH TOKENIZATION AND DETOKENIZATION

Aiming for enhanced audio understanding, we utilize Whisper-Large-v3 **Speech Encoder** to extract continuous audio representations at the user end. Subsequently, an **Adapter** is introduced to downsample the temporal resolution of these representations and align their hidden dimension

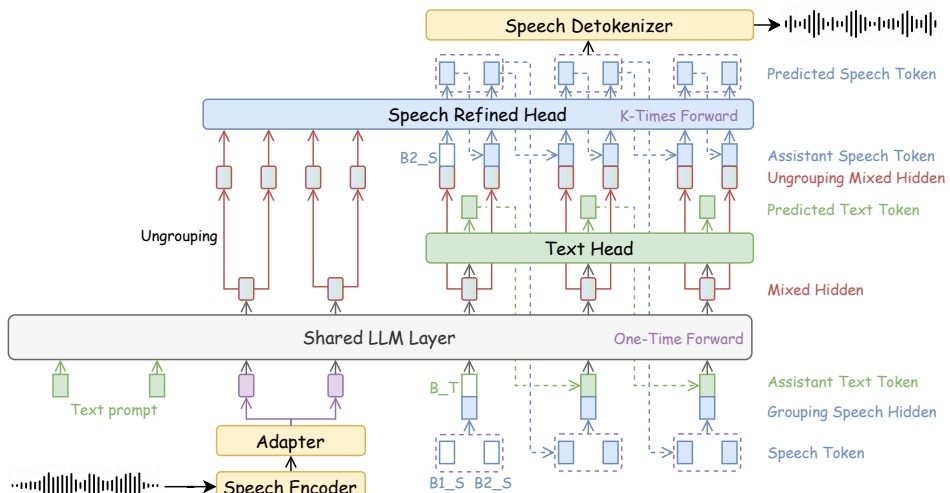

Figure 1: Overview of DRVOICE. User speech inputs are tokenized, *grouped*, and encoded by the MLLM for autoregressive text and speech token prediction. The MLLM consists of **Shared LLM Layer**, a **Text Head**, and a **Speech Refined Head (SRH)** for token generation. The generated speech tokens are then converted to speech waveform by the speech detokenizer. Note that SRH generates $k$ speech tokens through $k$ autoregressive forward passes, where $k$ is the grouping factor.

with that of the LLM. Semantic tokens have been widely used for speech tokenization (Zhang et al., 2023a; Borsos et al., 2023), due to their strong alignment with text (Zhang et al., 2023b); therefore, we employ S3Tokenizer (Du et al., 2024a;b; 2025) as the **Speech Tokenizer** to convert speech waveform to semantic speech token sequence $\mathbf{S} = [s_0, s_1, \cdots, s_{T-1}]$ at the assistant end, where $T$ is the speech token sequence length. For speech detokenization, conditioned on speaker embeddings capturing acoustic details such as timbre, the Flow Matching model (Lipman et al., 2023) converts speech tokens $\mathbf{S}$ into the Mel spectrum for a given speaker. Finally, a pre-trained vocoder HiFi-GAN (Kong et al., 2020) transforms the Mel spectrum back into audio signal.

## 3.2 MULTIMODAL LARGE LANGUAGE MODEL (MLLM)

Built upon text-LLMs, the MLLM learns *joint speech-text modeling* to process speech or text inputs while producing *parallel* speech and text outputs.

**Parallel Joint Speech-Text Model.** Inspired by Moshi (Défossez et al., 2024), explicit text streaming is incorporated to aid speech generation as a common semantic scaffold. We focus on performing modality alignment **exclusively at the assistant end**. *This design adheres to the **asymmetric** nature of human-machine interactions: while user inputs are typically unimodal (either text or speech), the assistant's responses can be a coordinated multimodal output.*

Leveraging the autoregressive generation capability of LLMs, both generated speech tokens $s_t$ and text tokens $t_t$ are iteratively fed back into the shared LLM layer at each timestep. Their embeddings are added as model input, forming a parallel speech-text architecture. Formally, the combined input embedding $c_t$ at timestep $t$ is computed as:

$$c_t = E_{\text{speech}}(s_t) + E_{\text{text}}(t_t) \tag{1}$$

where $E_{\text{speech}}$ and $E_{\text{text}}$ denote the embeddings of speech and text tokens, respectively. Since the lengths of speech tokens and text tokens are typically mismatched, the shorter sequence is padded with a special token <|SIL|>to align them for each utterance. The autoregressive generation process is as follows:

$$P(y_t|y_{<t}, x) = \prod_{i=1}^{t} P(y_i|y_{<i}, x) \tag{2}$$

where $x$ is the input sequence and $y_t = (s_t, t_t)$ denotes the joint speech-text output at timestep $t$. This unified formulation enables seamless integration of speech and text generation within a *single* autoregressive framework. To preserve the intrinsic linguistic understanding and generation capabilities of pretrained text-LLMs while enabling cross-modal interactions, three key methodological designs are introduced for intermodal coordination, including Speech Token Grouping and Ungrouping, and Speech Refined Head.

**Speech Token Grouping.** A crucial parameter for discrete tokens is the sampling rate, which determines the input/output sequence length. GLM-4-Voice (Zeng et al., 2025) investigates sampling rates from 6.25Hz to 50Hz on LibriSpeech, revealing minimal differences in Word Error Rate (WER) between 50Hz and 25Hz but significant degradation at 12.5Hz and 6.25Hz. Hence, our work adopts 25Hz sampling rate. To resolve the temporal resolution mismatch between speech signals (25Hz) and text tokenization ($\sim$3Hz) (Chen et al., 2024a), a **grouping** mechanism is designed:

$$\mathbf{g}_i = \mathrm{Linear}\left( \overset{(i+1)k-1}{\underset{j=ik}{\|}} \mathbf{s}_j \right) \in \mathbb{R}^{d_{\text{text}}} \tag{3}$$

where $\mathbf{s}_j$ denotes speech tokens, $\|$ represents feature concatenation, and $k$ is the grouping factor determined by the ratio between speech token rates and text token rates. This design compresses the speech token length from $T$ to $T/k$ and the resulting grouped speech representations align better with text. Notably, different from Chen et al. (2024a) that employs linear projection of audio logits into group-sized representations for *parallel multi-token prediction*, DRVOICE strategically designs an **ungrouping** mechanism and incorporates a dedicated Speech Refined Head (SRH) to enable autoregressive generation of *individual* speech tokens.

**Speech Refined Head (SRH).** SRH is proposed to enhance speech generation capabilities. It utilizes the last hidden state of the shared LLM layer (SLLM) as conditional input and incorporates contextual speech information to autoregressively generate speech tokens. While conventional speech grouping strategies–which cluster speech tokens into semantically meaningful units–have proven effective for speech recognition and understanding tasks (Chen et al., 2024a; Zhang et al., 2024b), our experiments reveal their inherent limitations in *generative* scenarios, since speech token grouping inevitably loses some fine-grained acoustic details. To address this limitation, DRVOICE performs an **ungrouping** process as follows: the SLLM's final hidden state is mapped to group-sized embeddings via linear projection

$$\mathbf{h}_{ug} = \mathbf{W}_p \mathbf{h}_L^{[\text{SLLM}]} \quad \text{where} \quad \mathbf{W}_p \in \mathbb{R}^{d_g \times d_h}, \tag{4}$$

and time splitting

$$\mathbf{H} = \mathrm{Split}_k(\mathbf{h}_{ug}) = [\mathbf{h}_{ug}^{(1)}, \mathbf{h}_{ug}^{(2)}, \ldots, \mathbf{h}_{ug}^{(k)}], \tag{5}$$

where $\mathbf{h}_{ug}^{(i)} \in \mathbb{R}^{d_{ug}/k}$. The resulting $\mathbf{H}$ serves as the conditional input for SRH that autoregressively generates speech tokens. Following our practice on SLLM, representations of preceding speech tokens and $\mathbf{H}$ are aggregated. Speech token prediction is trained to maximize the conditional probability:

$$\mathcal{L}_{\text{SRH}} = -\sum_{i=1}^{T} \log P(s_i | s_{<i}, \mathbf{H}_{<i}), \tag{6}$$

where $s_i$ represents the $i$-th speech token. By optimizing this objective, SRH learns to predict subsequent speech tokens conditioned on both preceding speech tokens and the rich contextual embeddings $\mathbf{H}$ derived from SLLM. This design enables SRH to effectively leverage the semantic and acoustic information encoded in the hidden representations of SLLM, thereby producing more natural and coherent speech outputs compared to conventional grouping-based approaches.

The E2E training objective $\mathcal{L}_{\text{MLLM}}$ integrates the two losses through multi-task learning:

$$\mathcal{L}_{\text{MLLM}} = \lambda \mathcal{L}_{\text{TH}} + \mu \mathcal{L}_{\text{SRH}}, \tag{7}$$

$$\mathcal{L}_{\text{TH}} = -\sum_{i=1}^{T} \log P(t_i | c_{<i}, \mathbf{g}), \tag{8}$$

where $\mathcal{L}_{\text{TH}}$ is the autoregressive loss over the text head, $\lambda$ and $\mu$ are hyperparameters.

Table 1: Multimodal Interaction Patterns.

| Pattern Name | Abbr. | Modality Flow |
|---|---|---|
| Speech-to-Multimodal | S2M | Speech → Joint speech-text response |
| Speech-to-Text | S2T | Speech → Text-only response |
| Text-to-Multimodal | T2M | Text → Joint speech-text response |
| Text-to-Text | T2T | Text → Text-only response |
| Speech-Text Chain | STC | Speech → Text transcription → Text response → Multimodal response |
| Speech-Assisted Chain | SAC | Speech → Text response (agent perspective) → Multimodal response |
| Speech-User Chain | SUC | Speech → Text transcription (user perspective) → Multimodal response |

## 3.3 TRAINING STRATEGY

**Initialization.** The Speech Encoder is initialized with the weights of Whisper-Large-v3, while the Shared LLM Layer is initialized using Qwen2.5. Additionally, the pre-trained Speech Tokenizer and Detokenizer from CosyVoice are employed and kept frozen throughout the entire training process. We initialize SRH with a pre-trained TTS model.

**CoM-Mixing Training.** The chain-of-modality (CoM) methodology (Zhang et al., 2023a) can enhance performance by leveraging intermediate textual transcriptions, which is particularly suitable for scenarios where real-time processing is not critical. Furthermore, practical applications involve scenarios where both speech and text output or text-only output are required, necessitating the system to dynamically generate different modalities based on specific needs. Seven interaction patterns for diverse input-output requirements are summarized in Table 1. System prompts are employed to regulate the model's output behavior. For example, prompts such as "`[System] You are a helpful assistant and asked to generate both text and speech tokens at the same time.`" explicitly guide the model to produce multimodal outputs. Detailed prompts are shown in Appendix B. During inference, specifying these system prompts enables the generation of desired output results. For CoM-Mixing training, we construct data variants following the seven interaction patterns and obtain a mixture of data for training the model. During inference, user can manually prepend the appropriate system prompt to the input sequence to meet the output requirement. This flexibility ensures adaptability to varying modality generation demands.

**Core-Cocktail Training.** We identify a learning rate dilemma: employing a high learning rate significantly compromises the performance of the MLLM, whereas using a low learning rate leads to training stagnation, with the loss decreasing quite slowly. To overcome this optimization challenge, we develop a specialized two-stage training strategy, termed "Core-Cocktail". The first stage involves full fine-tuning the entire MLLM with a relatively high learning rate. The goal of this phase is to rapidly move the model's parameters into a more favorable region of the loss landscape. However, to counteract the potential performance degradation of the MLLM caused by this aggressive initial training, an intermediate merging step is introduced. Drawing inspiration from Xiao et al. (2024), the parameters of the MLLM trained in the first stage ($M_1$) are merged with the parameters of the base, pre-trained LLM ($M_0$). This merging creates a new, interpolated model, $M_r$, as defined by the following equation:

$$M_r \leftarrow \alpha M_1 + (1 - \alpha)M_0$$

where $M_r$ denotes the merged model and $\alpha$ is the interpolation weight. The merging step effectively re-integrates the robust knowledge of the base LLM. A smaller $\alpha$ corresponds to a greater preservation of the base LLM's capabilities. The second stage conducts full fine-tuning on the merged model $M_r$ using a small learning rate. The core-cocktail training approach allows for careful and precise optimization, refining the model's performance without instability from a high learning rate.

## 4 EXPERIMENTS

### 4.1 EXPERIMENTAL SETUP

**Datasets.** Approximately 100K hours of audio-text paired data for speech-text modality alignment is adopted for pre-training Speech Refined Head. For DRVOICE post-training, we first synthesize speech for about 3B text tokens using CosyVoice (Du et al., 2024a;b; 2025), then select about 26K hours for

Table 2: Performance Comparison on various benchmarks in terms of benchmark-specific metrics (the best and second-best results in each row are in **bold** and underlined, respectively). With the exception of GLM4-Voice, whose results are cited from Xu et al. (2025) and Chen et al. (2024b), and the BBA results, which are cited from Xiaomi (2025) except for MiniCPM-o 2.6 that we reproduce, all other results were generated by running inference on the released checkpoints. **FR(In/Out)** denotes the input speech frame rate and the output speech plus text frame rate for the LLM backbone. $\tau$ denotes the average number of text tokens corresponding to one second of speech.

| | GLM4-Voice | MiniCPM -o 2.6 | Baichuan -Omni-1.5 | Qwen2.5 -Omni | Kimi -Audio | Step-Audio2 -Mini | DRVOICE |
|---|---|---|---|---|---|---|---|
| FR (In/Out) | $12.5/12.5+\tau$ | $25/\tau$ | $12.5/12.5+\tau$ | $25/\tau$ | $12.5/12.5$ | $12.5/25+\tau$ | **5/5** |
| *OpenAudioBench (S2T)* | | | | | | | |
| AlpacaEval | 57.89 | 64.10 | 77.90 | 72.76 | 75.73 | 59.60 | **82.61** |
| Llama Q. | 76.00 | 78.00 | 78.50 | 75.33 | 79.33 | 75.00 | **83.00** |
| Reasoning QA | 47.43 | 38.60 | 50.00 | **63.76** | 58.02 | 46.04 | 59.90 |
| TriviaQA | 51.80 | 63.00 | 57.20 | 57.06 | 62.10 | 57.70 | **64.50** |
| Web Q. | 55.40 | 69.20 | 59.10 | 62.80 | **70.20** | 65.10 | **70.20** |
| Overall | 57.70 | 62.58 | 64.54 | 66.34 | 69.08 | 60.69 | **72.04** |
| *VoiceBench (S2T)* | | | | | | | |
| AlpacaEval | 3.97 | 4.42 | 4.50 | 4.33 | 4.46 | 4.17 | **4.74** |
| CommonEval | 3.42 | **4.15** | 4.05 | 3.84 | 3.97 | 3.00 | 4.08 |
| SD-QA | 36.98 | 50.72 | 43.40 | 57.41 | 63.12 | 56.06 | **64.30** |
| MMSU | 39.75 | 54.78 | 57.25 | 56.38 | 62.17 | 52.18 | **67.27** |
| OpenBookQA | 53.41 | 78.02 | 74.51 | 79.12 | **83.52** | 64.18 | 82.20 |
| IFEval | 52.80 | 49.25 | 54.54 | 53.88 | 61.10 | 38.01 | **71.39** |
| AdvBench | 88.08 | 97.69 | 97.31 | 99.62 | **100.00** | 93.08 | 99.62 |
| Overall | 59.83 | 71.69 | 71.14 | 72.83 | 76.93 | 63.84 | **80.17** |
| *UltraEval-Audio (S2S)* | | | | | | | |
| AlpacaEval | 51.00 | 51.00 | 58.69 | 56.10 | 44.20 | 51.72 | **59.29** |
| Llama Q. | 50.00 | 61.00 | 67.33 | 66.30 | 57.33 | 67.67 | **75.33** |
| TriviaQA | 36.40 | 40.20 | 30.57 | 40.52 | 35.71 | 33.50 | **46.09** |
| Web Q. | 32.00 | 40.00 | 38.09 | 38.93 | 33.90 | 34.65 | **45.92** |
| Overall | 42.35 | 48.05 | 48.67 | 50.46 | 42.79 | 46.89 | **56.66** |
| *Big Bench Audio (S2T & S2S)* | | | | | | | |
| S2T | 44.8 | 56.2 | 47.1 | 54.2 | 59.4 | 50.9 | **76.9** |
| S2S | 42.7 | 55.4 | 44.6 | 53.6 | 51.0 | 47.5 | **71.2** |
| Overall | 43.8 | 55.8 | 45.8 | 53.9 | 55.2 | 49.2 | **74.0** |

Table 3: Speech quality performance on the collections of UltraEval-Audio, in terms of UTMOS for assessing overall speech quality and ASR-WER for assessing alignment between generated speech and text. **FR(In/Out)** indicates the input speech frame rate and the output (speech + text) frame rate for the LLM backbone, while $\tau$ represents the average text tokens per second of speech.

| Model | FR(In/Out)↓ | UTMOS↑ | ASR-WER↓ |
|---|---|---|---|
| MiniCPM-o 2.6 (2025) | $25/\tau$ | 4.18 | 13.17 |
| Baichuan-Omni-1.5 (2025) | $12.5/12.5+\tau$ | 4.27 | 23.38 |
| Qwen2.5-Omni (2025) | $25/\tau$ | 4.28 | **3.48** |
| Kimi-Audio (2025) | $12.5/12.5$ | 3.06 | 21.06 |
| Step-Audio2-mini (2025) | $12.5/25+\tau$ | **4.53** | 9.50 |
| **DRVOICE** | **5/5** | 4.29 | 8.36 |

speech-to-speech conversation and about 20K hours user speech plus 1.3B assistant tokens for speech-to-text conversation, based on Word Error Rate (WER) of the synthesized speech. Furthermore, to enhance the model's comprehension of real-world speech, the training data was augmented with about 10K hours of English Automatic Speech Recognition (ASR) data, comprising several corpora, including Common Voice (Ardila et al., 2020), MELD (Poria et al., 2019), LibriSpeech (Panayotov et al., 2015), SPGISpeech (O'Neill et al., 2021), and Voxpopuli (Wang et al., 2021). Following prior works (Yao et al., 2024; KimiTeam et al., 2025; OpenAI, 2024b), we evaluate the performance on the

Table 4: Ablation study of Continuous Speech Encoder (CSE), Speech Refined Head (SRH), SRH-Pretraining and CoM-Mixing on Llama Questions. **S2M, S2T, etc** are defined in Table 1.

| Model | S2M (T/S) | S2T | T2M (T/S) | T2T | STC (T/S) | SAC (T/S) | SUC (T/S) |
|---|---|---|---|---|---|---|---|
| DRVOICE-Small | 68.67 / 56.00 | 72.33 | 72.33 / 56.00 | 75.33 | 75.67 / 68.33 | 71.67 / 62.67 | 73.33 / 62.00 |
| w/o. CSE | 61.67 / 53.00 | 62.33 | 70.00 / 60.00 | 74.00 | 69.33 / 61.00 | 63.00 / 55.00 | 66.33 / 58.67 |
| w/o. SRH-Pretraining | 38.33 / 30.33 | 56.00 | 59.33 / 46.33 | 73.33 | 67.33 / 57.67 | 54.00 / 42.33 | 54.33 / 42.67 |
| w/o. SRH | 21.67 / 15.33 | 56.00 | 45.22 / 35.00 | 73.00 | 64.33 / 50.67 | 55.67 / 42.33 | 40.33 / 27.67 |
| w/o. CoM-Mixing | 58.00 / 49.00 | 58.00 | 69.33 / 55.00 | 68.33 | –/– | –/– | –/– |

widely used benchmarks, VoiceBench (Chen et al., 2024b) and OpenAudioBench [2] for **Speech-To-Text** ($S \rightarrow T$) **evaluation**, UltraEval-Audio [3] for **Speech-to-Speech** ($S \rightarrow S$) **evaluation**, and Big Bench Audio [4] for both evaluations.

**Evaluation Metrics.** Evaluations adhere to the established protocols for each respective benchmark. Specifically, for the open-ended QA tasks on AlpacaEval and CommonEval, G-Eval (Liu et al., 2023) is used for scoring. For AdvBench, the Refusal Rate is reported, while performance on all other benchmarks is assessed with Accuracy. The generated speech is transcribed using Whisper-v3-large model (Radford et al., 2022), then WER (denoted by ASR-WER) of transcripts is computed against the generated text to assess the alignment between generated speech and text. UTMOS (Saeki et al., 2022) is used to evaluate the overall speech quality, following Zeng et al. (2025).

**Baselines.** Representative and competitive open-source audio language models are selected as baselines to cover diverse modeling paradigms: Text-Driven models including MiniCPM-o 2.6 (8B) (Yao et al., 2024) and Qwen2.5-Omni (7B) (Xu et al., 2025). Among Joint Speech-Text models, interleaved models including GLM-4-Voice (9B) (Zeng et al., 2025), Baichuan-Omni-1.5 (7B) (Li et al., 2025), and Step-Audio2-Mini (8B) (Wu et al., 2025), alongside the parallel model Kimi-Audio (7B) (KimiTeam et al., 2025). This suite enables systematic comparisons across mainstream speech-text modeling strategies.

**Implementation Details** can be found in Appendix A.

## 4.2 MAIN RESULTS

**Overall Performance.** Table 2 compares DRVOICE (7B) with representative and competitive baselines on speech-to-text (S→T) and speech-to-speech (S→S) generation performance. As shown in the table, DRVOICE demonstrates exceptional capabilities across a wide range of tasks, **achieving new state-of-the-art (SOTA) results on all four major benchmarks**. Specifically, DRVOICE secures the top position on **OpenAudioBench** (audio understanding) with an overall score of **72.04**, on **VoiceBench** (benchmarking LLM-based voice assistants) with **80.17**, on **UltraEval-Audio** (both speech understanding and generation) with **56.66**, and on **Big Bench Audio** (reasoning and understanding) with **74.0**. *This consistently dominant and balanced performance across diverse benchmarks establishes DRVOICE as the leading open-source speech foundation model in the ~7B parameter class.*

**Computational Efficiency and Speech Quality.** A key innovation of DRVOICE is its remarkable computational efficiency, highlighted in Table 2 and Table 3. As shown in the **FR (In/Out)** rows, DRVOICE operates at a frame rate of **5/5**, indicating that the LLM backbone processes only 5 audio tokens per second for both input and output. This is a substantial reduction compared to other models, which operate at frame rates of 12.5 or 25, thereby significantly lowering computational requirements and potential latency. Crucially, this efficiency does not come at the cost of speech quality. Table 3 shows that despite its low frame rate, DRVOICE produces high-quality and well-aligned speech. It achieves UTMOS score of **4.29** for overall speech quality, which is competitive with top-performing models like Qwen2.5-Omni (4.28) and superior to others like Kimi-Audio (3.06). With an ASR-WER of 8.36, the model shows robust speech-text alignment and surpasses several baselines. Still, it underperforms compared to Qwen2.5-Omni. A potential explanation lies in the architectural design:

---

[2] https://huggingface.co/datasets/baichuan-inc/OpenAudioBench
[3] https://github.com/OpenBMB/UltraEval-Audio
[4] https://huggingface.co/datasets/ArtificialAnalysis/big_bench_audio

Qwen2.5-Omni's feeding text directly to its "Talker" module, whereas the proposed model only sends hidden states to SRH. **This unique combination of SOTA performance, high-quality speech generation, and unparalleled efficiency makes DRVOICE a highly powerful and practical model for real-world applications**.

## 4.3 ABLATION AND ANALYSES

We conduct extensive ablation study and analyses. For computational efficiency, some experiments are performed on DRVOICE-Small (1.5B). More analyses about the data quality and data scaling can be found in Appendix C.

**Core-Cocktail Training Strategy.** While Stage 1 training causes a performance drop from text baseline of 81.77 to 70.19, Stage 2 reverses this trend, recovering the performance to 74.73. This result confirms that the strategy effectively counteracts the initial degradation and leads to an optimized model. Further details are available in Appendix C.

**Continuous Speech Encoder.** As shown in Table 4, the Continuous Speech Encoder (CSE) is vital for tasks involving speech inputs. In this setting, user audio input is processed solely using the discrete speech tokenizer (same as the assistant's output tokenizer), without the Whisper features. Removing it from DRVOICE-Small leads to a significant performance drop in speech understanding and speech generation. Specifically, the S2T score decreases from 72.33 to 62.33 (13.8% relatively), and the S2M (T) score falls from 68.67 to 61.67 (10.2% relatively). In contrast, the impact on text-only tasks is minimal, with the T2T score only slightly decreasing from 75.33 to 74.00. This confirms the CSE's effectiveness in representing speech.

**Dual-Resolution Speech Representations (DRSR).** Our dual-resolution approach combines input grouping for efficiency and comprehension with a refinement head for high-quality generation. **1) Grouping Factor.** Contrary to degrading generation, grouping substantially improves both speech understanding (S2T) and speech-to-speech generation (S2M). For instance, increasing the grouping factor from 1 to 5 raises the S2T score from 55.67 to 63.33 (a 13.7% relative gain). The S2M (T/S) score sees a significant improvement, peaking at 37.67 / 28.00 with a grouping factor of 5. Furthermore, using a grouping factor of 5 instead of 1 reduces nearly 50% GPU hours in each setting, proving the efficiency of grouping machinism. Detail results can be found in Appendix C. **2) Speech Refinement Head.** As shown in Table 4, the Speech Refinement Head (SRH) is remarkably effective for speech token generation tasks. To handle the resolution mismatch when SRH is removed, the hidden state from the shared LLM is passed through a projection layer ($d_{model} \rightarrow k \times d_{speech\_token}$) and then split into k tokens. This setup simulates a standard parallel prediction approach without autoregressive refinement. By comparing the model with SRH (w/o. SRH-Pretraining) to one without (w/o. SRH), we observe that adding SRH achieves a 76.9% relative improvement in S2M (T) (from 21.67 to 38.33) and a 31.2% relative improvement in T2M (T) (from 45.22 to 59.33). The text-only performance (T2T) remains stable, confirming that SRH enhances speech generation without interfering with text processing pathways. Our dual-resolution architecture effectively combines these components, using grouping for efficient comprehension and SRH at the raw frame resolution for speech generation.

**SRH-Pretraining.** To examine the impact of pretraining, we remove the SRH pretraining step and retrain the model. As shown in Table 4, removing pretraining (comparing w/o. CSE with w/o. SRH-Pretraining) has the most significant impact on speech generation, causing S2M (T) performance to drop by 37.8% and T2M (T) by 15.2%. The effect on S2T is smaller (10.2% drop), and negligible for T2T. This underscores the critical importance of SRH pretraining for refining speech token generation.

**CoM-Mixing Training Strategy.** As shown in Table 4, where tasks guided by contextual system prompts (STC/SAC/SUC) demonstrate substantial improvements over direct S2M generation. For example, STC (T) achieves a score of 75.67, significantly surpassing the S2M (T) baseline of 68.67. This shows the model successfully learns to adopt the generation paradigm guided by system prompts. The data augmentation effect is confirmed by ablating the CoM-Mixing strategy entirely. Training without it (w/o. CoM-Mixing Training) results in a 15.5% relative performance degradation in S2M (T) (from 68.67 to 58.00), confirming the value of our mixed-modality training approach.

## 5 CONCLUSIONS

We introduce DRVOICE, a novel parallel speech-text voice conversation model that leverages joint autoregressive modeling with dual-resolution speech representations. Our experimental results demonstrated that DRVOICE establishes a new state-of-the-art (SOTA) on OpenAudioBench for audio understanding, VoiceBench for voice assistant tasks, UltraEval-Audio for both speech understanding and generation and on Big Bench Audio for reasoning capabilities, confirming its strong and versatile capabilities. Notably, its dual-resolution mechanism significantly improves inference speed and computational efficiency (empirically achieving a reduction of nearly 50% in GPU hours for training) without sacrificing performance. We discuss limitations and future work in Appendix D.

## REPRODUCIBILITY STATEMENT

To ensure the reproducibility of our results, we will make our resources publicly available. The complete source code for our model, training and evaluation scripts, and all pre-trained model checkpoints will be released upon publication. The speech data used in our experiments were synthesized using the publicly available CosyVoice model; we will provide the necessary scripts and instructions to replicate the dataset. Furthermore, a comprehensive description of the implementation details, including architectural choices, hyperparameters, and training setup, is provided in Appendix A. We believe these resources are sufficient for the community to reproduce our findings and build upon our work.

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

# Appendices

## A   IMPLEMENTATION DETAILS

Qwen2.5-0.5B is trained following $T2M$ paradigm with Speech-text Alignment data to initialize SRH (SRH-PT). The maximum sequence length is set to 2K tokens, which corresponds to an audio duration of approximately 6.8 minutes. DRVOICE uses Qwen2.5-7B-Instruct as its base LLM, while DRVOICE-Small utilizes Qwen2.5-1.5B-Instruct as the base LLM. Grouping factor is set to $k = 5$. Core-cocktail interpolated factor is set to $\alpha = 0$ for extremely maintaining the base LLM capability. Multiple loss hyperparameters are set to $\lambda = 1$ and $\mu = 1$. The warmup rate is set to $2\%$ of the total training steps. For the two-stage training of DRVOICE, the learning rate is decayed from $1 \times 10^{-4}$ to $1 \times 10^{-5}$ in stage one, and subsequently from $2 \times 10^{-5}$ to $2 \times 10^{-6}$ in stage two, with both stages utilizing a cosine annealing schedule. In contrast, DRVOICE-Small and SRH-PT are trained in a single stage, which adopts the same learning rate schedule as the first stage of DRVOICE. The AdamW method (Loshchilov & Hutter, 2019) is used for optimization. Experiments are run on 64x NVIDIA Tesla A800 80G GPUs with Brain floating-point format (BF16) (Kalamkar et al., 2019) to accelerate training and decoding. For DRVOICE, DeepSpeed ZeRO-2 (Rajbhandari et al., 2020) is implemented to prevent GPU out-of-memory. It takes about 20 hours on SRH-PT, and about 45 hours on DRVOICE post-training.

## B   PROMPT TEMPLATES

**System prompts for multimodal interaction patterns.**   As shown in Table 5, there are five types of system prompts categorized based on the model's output partitioning. During inference, specifying these system prompts enables the generation of desired output results.

Table 5: System Prompts for Multimodal Interaction.

| Pattern Abbr. | System Prompts |
|---|---|
| S2M & T2M | You are a helpful assistant and asked to generate both text and speech tokens at the same time. |
| S2T & T2T | You are a helpful assistant and asked to generate text tokens. |
| STC | You are a helpful assistant. Let's think step by step. Convert speech to text if the query is speech, think of an appropriate text response, and then convert the response back to both text and speech tokens at the same time. |
| SAC | You are a helpful assistant. Let's think step by step. Think of an appropriate text response, and then convert the response back to both text and speech tokens at the same time. |
| SUC | You are a helpful assistant. Let's think step by step. Convert speech to text if the query is speech, and then think of both appropriate text and speech responses at the same time. |

## C   MORE ABLATION AND ANALYSES

**Core-Cocktail Training Strategy.** Table 6 shows the precise trade-offs and benefits of our two-stage approach. For instance, under the comprehensive *Inhouse v2, MagpiePro, InfGen* dataset setting, the model's average performance after Stage 1 drops to 70.19, a significant decline of over 11 points compared to the text-only baseline (81.77). This observation perfectly aligns with our initial hypothesis: the aggressive full fine-tuning with a high learning rate in Stage 1, while intended to rapidly move parameters into a more favorable region of the loss landscape, temporarily compromises the model's general capabilities, as predicted. This performance degradation is precisely the problem that the subsequent steps are designed to rectify. Following the intermediate merging step—which "cocktails" the aggressively trained model with the base LLM to re-integrate its robust

Table 6: Training Strategy Performance comparison on the Voicebench benchmark with different training datasets.

| Description | AlpacaEval | CommonEval | SD-QA | MMSU | OpenBookQA | IFEval | AdvBench | Avg |
|---|---|---|---|---|---|---|---|---|
| Qwen2.5-7B-Instruct (T2T) | 4.67 | 4.34 | 76.13 | 69.97 | 82.20 | 65.45 | 99.04 | 81.86 |
| *Inhouse v1* | | | | | | | | |
| OnlyText-Training (T2T) | 4.15 | 3.57 | 60.15 | 54.50 | 66.50 | 50.22 | 99.40 | 69.31 |
| Stage 2 (S2T) | 4.08 | 3.54 | 55.88 | 52.80 | 69.45 | 38.48 | 99.23 | 66.89 |
| *Inhouse v2* | | | | | | | | |
| OnlyText-Training (T2T) | 4.22 | 3.60 | 68.35 | 58.10 | 68.13 | 65.69 | 99.62 | 73.76 |
| Stage 2 (S2T) | 4.10 | 3.15 | 64.01 | 58.30 | 79.56 | 55.25 | 98.27 | 71.48 |
| *Inhouse v2, MagpiePro, InfGen* | | | | | | | | |
| OnlyText-Training (T2T) | 4.64 | 4.26 | 71.79 | 69.97 | 84.40 | 68.84 | 99.42 | 81.77 |
| Stage 1 (S2T) | 4.25 | 3.09 | 57.32 | 54.29 | 75.82 | 58.07 | 99.04 | 70.19 |
| Stage 2 (S2T) | 4.54 | 3.35 | 64.56 | 61.61 | 80.44 | 59.83 | 98.85 | 74.73 |
| Stage 2 w/. ASR data (S2T) | 4.52 | 3.77 | 68.54 | 60.31 | 79.56 | 59.30 | 98.65 | 76.02 |

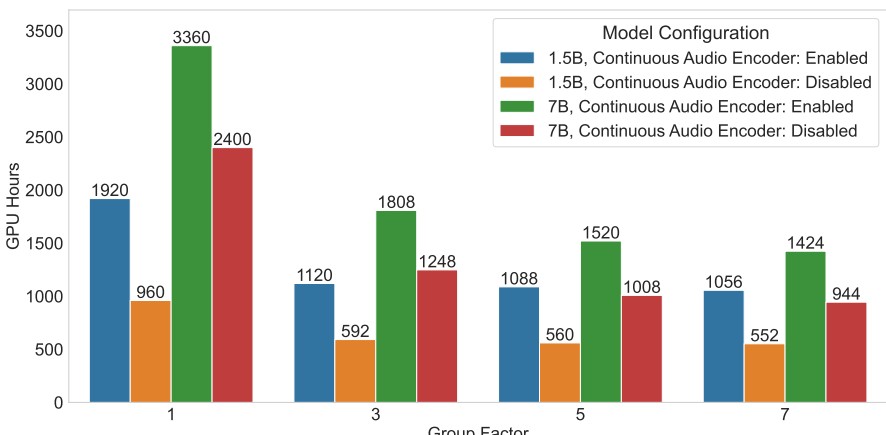

Figure 2: Computational Resources under 17K hours training data across different Grouping Factor.

knowledge—Stage 2 proceeds with careful fine-tuning using a small learning rate. As shown in the table, this second stage successfully recovers and refines the model, boosting the average score significantly to 74.73. This substantial improvement from Stage 1 demonstrates that our strategy effectively mitigates the instability caused by the initial high-learning-rate training. This two-stage process confirms that the Core-Cocktail strategy provides a powerful solution to the learning rate dilemma, enabling rapid adaptation without sacrificing final performance.

**Data Quality.** The *OnlyText-Training (T2T)* results in Table 6 serve as a powerful indicator of the maximum potential embedded within the textual content of our datasets. There is a clear correlation between data quality and the final model's performance: as the T2T average score improves from *Inhouse v1* (69.31) to our most comprehensive dataset (81.77), the final *Stage 2 (S2T)* model performance also sees a substantial lift. Notably, our final dataset mixture enables the T2T model to reach an average score of 81.77, almost perfectly matching the text-only Qwen2.5-7B-Instruct backbone. This confirms the exceptional quality of our instructional data and establishes a high-performance ceiling. Our final S2T model (74.73) successfully translates a significant portion of this potential into the multimodal domain.

Furthermore, the results underscore the critical importance of data composition for specific tasks. The CommonEval benchmark, which evaluates models on real human speech, is a case in point. Our standard *Stage 2 (S2T)* model achieves a score of 3.35 on this challenging benchmark. However, by augmenting the training data with ASR-derived pairs (*Stage 2 w/. ASR data*), the CommonEval score noticeably increases to 3.77. This gain is because ASR data exposes the model to the natural

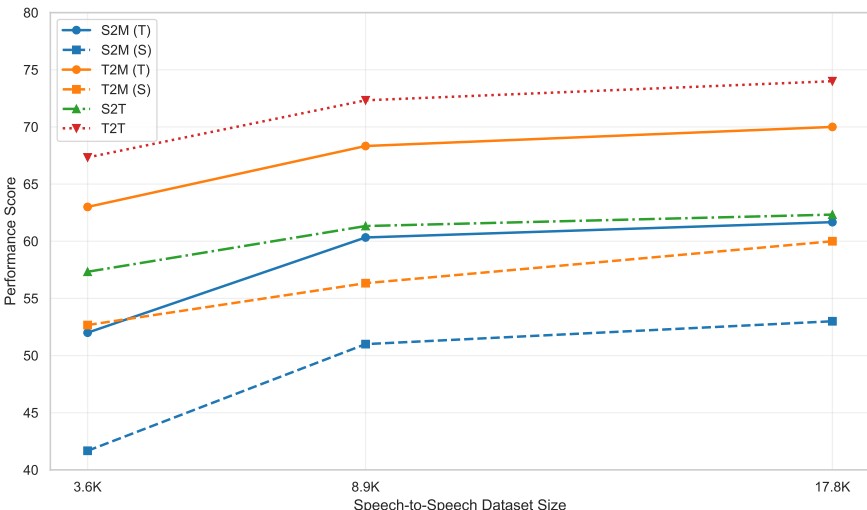

Figure 3: Performance Scaling of DRVOICE-Small (w/o. Continuous Speech Encoder) on LLaMA Question Benchmark.

disfluencies, varied intonations, and ambient noise present in real-world speech, enhancing its robustness.

**SRH-Pretraining Policy.** Two pre-training strategies were employed for the Speech Refined Head (SRH). The first strategy involves independently training a Text-to-Speech (TTS) task on the `Qwen2.5-0.5B` model in a T2M format with 25Hz token rate; the resulting trained weights are then loaded onto the SRH. In contrast, the second strategy freezes the Speech Encoder, Adaptor, Shared LLM Layer, and the Text Head, while exclusively training the SRH with a streaming TTS objective. Experiments demonstrated that the second strategy yields a substantial improvement in Speech-to-Speech (S2S) performance on the `UltraEval-Audio` benchmark, with the score increasing from 47.66 to 56.66. This significant gain is attributed to the closer alignment between the pre-training task in the second strategy and the format of the downstream task.

**Reinforcement learning.** To further enhance the model's capabilities in real-world speech comprehension, instruction following, and reasoning, reinforcement learning (RL) methods were employed. Specifically, to address the scarcity of authentic speech data, a preference dataset was constructed using open-source ASR datasets. A text LLM was used to identify samples suitable for user queries, for which canonical responses were then generated by another text LLM. Subsequently, the `DrVoice-SFT` model performed inference on the original speech queries to establish the preference data. For the instruction-following capability, the open-source preference dataset [5] was utilized. Training for these first two objectives was performed using the DPO (Rafailov et al., 2023) algorithm. Finally, to improve reasoning ability, the model underwent GSPO (Zheng et al., 2025) training on mathematical problems [6].

**Data Scaling.** To investigate the impact of data scaling, we conduct experiments by progressively expanding the S2S training data on DRVOICE-Small (w/o. Continous Speech Encoder). Each S2S instance can be augmented into 7 distinct multimodal interaction patterns. As illustrated in Figure 3, expanding the dataset from 3.6K to 17.8K samples yields consistent improvements. The performance of T2M(S) exhibits nearly linear growth, while other patterns, despite showing slower growth rates, still demonstrate upward trends in the figure, suggesting potential for further performance enhancement through additional data.

**Grouping Factor.** Table 7 and Figure 2 demonstrate that **our grouping strategy significantly enhances both performance and computational efficiency**. Contrary to degrading generation,

---

[5] `https://huggingface.co/datasets/allenai/tulu-3-pref-personas-instruction-following`

[6] `https://github.com/openai/prm800k`

Table 7: Impact of the Grouping Factor on Llama Q. S2T and S2M performance, using DRVOICE-Small (1.5B) w/o CSE and data subsets for faster experiments. The S2T model is trained exclusively on mixed S2T and T2T data, while the S2M model is trained on mixed S2M and T2T data.

| Grouping Factor | S2T | S2M (T/S) |
|---|---|---|
| 1 | 55.67 | 4.00 / 2.67 |
| 3 | 64.67 | 15.67 / 5.00 |
| 5 | 63.33 | 37.67 / 28.00 |
| 7 | 62.67 | 36.00 / 16.67 |

Table 8: Benchmark Results: VoiceBench (T2T)

| Data | VoiceBench | | | | | | | |
|---|---|---|---|---|---|---|---|---|
| | AlpacaEval | CommonEval | SD-QA | MMSU | OBQA | IFEval | AdvBench | Overall |
| Qwen2.5-7B-Instruct | 4.67 | 4.34 | 76.13 | 69.97 | 82.20 | 65.45 | 99.04 | 81.86 |
| Inf7M | 3.83 | 3.50 | 69.98 | 55.43 | 71.87 | 48.79 | 94.42 | 69.58 |
| InfGen | 4.51 | 4.13 | 74.14 | 62.46 | 76.04 | 58.35 | 98.46 | 77.46 |
| openorca_gpt4 | 4.33 | 4.03 | 73.96 | 61.00 | 73.85 | 54.05 | 96.35 | 75.20 |
| openorca_gpt3_5 | 4.35 | 4.17 | 71.07 | 57.84 | 66.59 | 49.39 | 95.00 | 72.90 |
| Evol_Instruct_Code | 4.19 | 4.06 | 69.26 | 63.14 | 81.54 | 56.40 | 90.38 | 75.10 |
| Evol_Instruct | 4.09 | 3.91 | 71.61 | 60.38 | 72.31 | 54.02 | 98.27 | 73.80 |
| magpie_pro_llama3_1_300k | 4.42 | 4.08 | 72.69 | 58.33 | 74.95 | 56.08 | 86.73 | 74.11 |
| magpie_pro_mt_llama3_1_300k | 4.46 | 4.05 | 73.60 | 58.07 | 60.00 | 60.05 | 86.35 | 72.61 |
| numinamath_cot | 4.11 | 3.78 | 70.16 | 55.43 | 68.35 | 52.00 | 97.69 | 71.63 |
| OpenHermes_v2_5 | 4.05 | 3.71 | 71.61 | 62.20 | 74.07 | 51.59 | 82.88 | 71.08 |
| Synthia_v1_3 | 4.30 | 3.96 | 72.33 | 63.63 | 76.70 | 53.32 | 97.69 | 75.55 |
| tulu_3_sft | 4.04 | 3.73 | 65.64 | 61.74 | 79.78 | 61.78 | 100.00 | 74.91 |
| magpie_pro_llama3_3_500k_filtered | 4.48 | 4.17 | 73.42 | 64.31 | 81.32 | 64.81 | 92.88 | 78.53 |

grouping substantially improves both speech understanding (S2T) and speech-to-speech generation (S2M). For instance, increasing the grouping factor from 1 to 5 raises the S2T score from 55.67 to 63.33 (a 13.7% relative gain). The S2M (T/S) score sees a significant improvement, peaking at 37.67 / 28.00 with a grouping factor of 5. Furthermore, as shown in Figure 2, using a grouping factor of 5 instead of 1 reduces nearly 50% GPU hours in each setting, proving the efficiency of grouping machinism.

**Retention vs Forgetting.** An attempt is made to select suitable open-source datasets for speech synthesis. All results are from fine-tuning Qwen2.5-7B-Instruct at a low learning rate (1e-5), so they primarily indicate retention vs. forgetting rather than absolute capability gains. The gap to the baseline quantifies forgetting: magpie_pro_llama3_3_500k_filtered and InfGen exhibit the smallest overall deltas (about 3–4 points on both suites), while very large, noisier corpora like Inf7M induce substantial forgetting despite their size. Quality and filtering matter more than scale: openorca_gpt4 forgets less than openorca_gpt3_5 with fewer tokens, and the magpie_pro family maintains instruction-following and factual QA best per token. Forgetting is selective: some submetrics improve (e.g., LlamaQ rises for Inf7M/tulu_3_sft; AdvBench reaches 100 for tulu_3_sft), but these gains often trade off against instruction and QA scores. Overall, compact, well-filtered, model-proximal data minimizes destructive drift at 1e-5, whereas bulk token count tends to amplify forgetting, so the most stable recipes preserve the base model while making targeted adjustments.

## D LIMITATIONS AND FUTURE WORK

Our future work will address limitations of this work and advance two key areas: (1) Enabling Full-Duplex Interaction: A crucial direction is to enable full-duplex interaction for more natural conversations. Inspired by recent advancements like Parrot (Wang et al., 2025), future work will investigate the use of a time-division multiplexing (TDM) input stream. This would permit DRVOICE to accept user speech inputs during its own speech generation phase, thereby efficiently utilizing idle time slots and allowing for responsive, interruptible dialogue. (2) Expanding to General Audio and Multimodality: Finally, a promising avenue is to extend the model's capabilities beyond speech-centric tasks to broader audio comprehension and synthesis,

Table 9: Benchmark Results: UltraEval-Audio (T2T) & OpenAudioBench (T2T)

| Data | Tokens (M) | UltraEval-Audio | | | | | OpenAudioBench | Total |
|---|---|---|---|---|---|---|---|---|
| | | AlpacaEval | LlamaQ | TriviaQA | WebQ | Overall | Reasoning QA | Avg |
| Qwen2.5-7B-Instruct | — | 81.41 | 82.33 | 53.91 | 52.56 | 67.55 | 58.00 | 75.10 |
| Inf7M | 2406 | 55.35 | 84.67 | 46.29 | 47.59 | 58.48 | 48.51 | 64.13 |
| InfGen | 1022.4 | 74.29 | 82.33 | 55.96 | 51.18 | 65.94 | 57.43 | 71.95 |
| openorca_gpt4 | 361.8 | 66.21 | 83.00 | 54.39 | 49.56 | 63.29 | 54.46 | 69.50 |
| openorca_gpt3_5 | 1101 | 55.91 | 83.00 | 54.69 | 48.47 | 60.52 | 48.51 | 66.74 |
| Evol_Instruct_Code | 28.3 | 52.32 | 81.33 | 52.64 | 44.24 | 57.63 | 52.48 | 67.39 |
| Evol_Instruct | 69.2 | 56.62 | 81.67 | 53.81 | 47.15 | 59.81 | 53.47 | 67.44 |
| magpie_pro_llama3_1_300k | 213.9 | 76.16 | 83.67 | 57.91 | 55.02 | 68.19 | 53.96 | 70.46 |
| magpie_pro_mt_llama3_1_300k | 323.2 | 75.35 | 83.00 | 54.00 | 54.53 | 66.72 | 57.43 | 69.38 |
| numinamath_cot | 463.2 | 64.04 | 80.67 | 54.88 | 50.64 | 62.56 | 50.00 | 66.81 |
| OpenHermes_v2_5 | 375.9 | 58.54 | 83.67 | 54.49 | 49.85 | 61.64 | 54.46 | 66.55 |
| Synthia_v1_3 | 59 | 61.67 | 83.67 | 53.81 | 47.88 | 61.76 | 55.45 | 69.28 |
| tulu_3_sft | 570.2 | 55.91 | 84.33 | 53.52 | 49.41 | 60.79 | 55.94 | 68.62 |
| magpie_pro_llama3_3_500k_filtered | 314.6 | 78.64 | 79.00 | 57.32 | 54.18 | 67.29 | 51.49 | 72.53 |

including the recognition and generation of music and environmental sounds. The subsequent integration of the visual modality will be a critical step toward developing a more comprehensive, multimodal conversational AI.

# E   THE USE OF LARGE LANGUAGE MODELS

During the preparation of this work, the authors used Large Language Models (LLMs) to improve grammar, clarity, and overall readability. The LLM was used as a writing aid and for language polishing purposes. The authors reviewed, edited, and take full responsibility for the final content and all claims presented in this manuscript.

