# OpenReview forum: "DrVoice: Parallel Speech-Text Voice Conversation Model via Dual-Resolution Speech Representations"
_ICLR.cc/2026/Conference — ICLR 2026 Poster_

### Official Review · Reviewer_mDyG · 2025-10-26

**Soundness:** 3
**Presentation:** 3
**Contribution:** 2
**Rating:** 4
**Confidence:** 3

**Summary:**

This paper proposes the use of dual speech representations (both continuous and discrete representations) and their grouping (to reduce the token rate and reduce the computation) as input along with the text tokens to an LLM and then at the output first a text head produces the text tokens which helps guide the speech head that produces the ungrouped sequence of speech tokens (referred to as Chain of Modalities). The output speech tokens are then converted into mel features via a flow matching model and a HiFiGAN vocoder is used to convert the mel features into the wave files. Hence, the system can take both speech and text as input and can produce output in both text and speech format (or a combination of the two). Based on the input output types, the paper describes 7 types of modes for the model which can be switched with a predetermined prompt. Another training detail that the paper emphasizes is that the importance of linearly mixing the base LLM's weights with that of the multimodal tuned model. Experiments on various speech+text benchmarks competitive performance to the several other multimodal LLMs including Qwen2.5-Omni  and Kimi-Audio. In addition, the paper claims to achieve the SOTA numbers on the Big Bench Audio tasks among the LLMs tested. In terms of generated speech quality, UTMOS score suggest that it is high quality audio even though the WER evaluation indicates some potential issues with the generated speech.

**Strengths:**

originality
* Use of both continuous and discrete representations of audio simultaneously is relatively less investigated in multimodal LLMs. Chain of modality and model weight mixing are presented as novelties which are somewhat novel. Especially, model averaging by linear combination is not particularly new but its application to LLMs during training might be an experimental design novelty.

quality
* The experimental results show competitive performance and the paper claims the SOTA results on the Big Bench Audio speech-to-text and speech-to-speech tasks.

clarity
* Mostly clearly written. However, some details are left to the Appendices.

significance:
* Successful implementation of multimodal LLMs is a relatively popular topic nowadays and this paper would be relevant to that community.

**Weaknesses:**

* The novelty is limited but sufficient (see the comments in the strengths section) Most of the individual components are well known (audio encoders, audio decoders, the LLM backbone, etc.)

* The main text skipped many details and provided those details in the Appendices. Which makes the reviewer questioning whether the paper needed more number of pages and a conference setting is not suitable for this paper.

* Even though one of the major claims of the paper is the computational cost reduction by grouping of audio tokens to reduce the token rate, the presented experiments do not provide the details of that improvement other than the token rate. What would be the computational savings in terms of FLOPs of RTF (whenever applicable)?

**Questions:**

1. Even though one of the major claims of the paper is the computational cost reduction by grouping of audio tokens to reduce the token rate, the presented experiments do not provide the details of that improvement other than the token rate. What would be the computational savings in terms of FLOPs of RTF (whenever applicable)?

2. Although the appendices contain some details, the main text seems to be glossing over the details. Do the authors think that the manuscripts would require a longer paper?

3. Based on the results in Table 2, the proposed DrVoice performs similarly to the other baselines in the table for all benchmarks except the Big Bench tasks. That is the relative differences are rather small between models. But for the Big Bench tasks we see a very large improvement (about 20% as compared to the Kimi Audio (the 2nd best)). Do the authors have any explanation of why DrVoice performs this well on these two tasks? Please comment on that.

**Details Of Ethics Concerns:**

Even though there is not a clear or explicit mention, it seems like the paper is a follow-up of the Kimi-Audio paper. Especially, the evaluations in Table 2, somewhat suggest that as the numbers are very close to each other in most cases.

---

> ### Author Response · Authors · 2025-11-27
> **Response to Reviewer mDyG [1/3]**
>
> We sincerely appreciate the Reviewer's thoughtful review and constructive feedback. Please first check our comment under "**Global Response to All Reviewers**". We hope the following point-by-point response could resolve all your remaining questions and concerns.
>
> > **Q1: Novelty of the approach.**
>
> Thank you for recognizing the novelty of our chain-of-modality training (our proposed **CoM-Mixing training**), model weight mixing of LLMs during training (part of our proposed **Core-Cocktail training**), and the combined use of continuous and discrete audio representations in the Strengths section. CoM-Mixing and Core-Cocktail Training are also recognized by **Reviewers 6YdV and EyEh** as strengths of our work.
>
> Importantly, we would like to emphasize that the **central innovation of this paper is the Dual-Resolution Architecture** (as highlighted by **Reviewers 6YdV and EyEh**).
>
> *   As Reviewer 6YdV acknowledged, "The **central idea of using dual-resolution speech representations is well-motivated** and **addresses a key, practical problem** in joint speech-text modeling: the significant discrepancy in token rates between speech and text. The proposed grouping/ungrouping mechanism is an **elegant solution** that directly tackles this issue, leading to **significant computational efficiency gains** (processing at 5Hz within the LLM) **without sacrificing output quality**, thanks to the Speech Refined Head.**"**
>
> *   Reviewer EyEh also recognized this core innovation as "speech token grouping benefits to reduce the computational costs and alleviate the frequency discrepancy between different representations. Leveraging the fine-grained acoustic information during generative scenarios is intuitive. The evaluation shows the speech quality is not compromised."
>
>
> As shown in Line 453-454 of the original submission, through the Dual-Resolution Architecture and the proposed CoM-Mixing and Core-Cocktail training strategies, DrVoice achieves nearly **50% reduction in GPU training time compared to no grouping** while maintaining SOTA performance (as shown in "**Global Response to All Reviewers**").
>
> > **Q2:** the main text seems to be glossing over the details; whether the paper needed more number of pages
>
> Thank you for the question. We respectfully disagree on the comment that "the main text seems to be glossing over the details" and your question on the structure of the original submission. We strongly believe that **our main text is self-contained, provides all essential information, and is well fit within the 9-page limit**.
>
> Specifically, **the 9-page main text is structured to be a complete and coherent presentation of our work**. The main text provides:
>
> 1.  **The motivation and background** for our research, positioning DrVoice in the context of existing voice conversation models.
>
> 2.  **The complete methodology**, including our primary architectural innovations: Dual-Resolution Speech Representations (DRSR), including Speech Token Grouping/Ungrouping, and the Speech Refined Head (SRH).
>
> 3.  **Our novel training strategies**, Core-Cocktail and CoM-Mixing, which are central to the model's performance.
>
> 4.  **The essential experiments**, including comparisons with representative and competitive baselines on the four major speech-to-text and speech-to-speech generation benchmarks (Table 2) and detailed analysis of Computational Efficiency and Speech Quality (Table 3) and the corresponding discussions.
>
> 5.  **The most critical ablation studies** (Table 4) that directly validate all of our core algorithmic designs, including Dual-Resolution Speech Representations and SRH, CoM-Mixing and Core-Cocktail training, and the continuous speech encoder.
>
>
> **In contrast, the Appendices** provide supplementary material that, while valuable for reproducibility and deeper analysis, is not essential for understanding the paper's primary claims. The Appendices include:
>
> 1.  **Implementation Details** such as hyperparameters, learning rate schedules, and hardware setup (Appendix A).
>
> 2.  **Detailed prompt templates** used for our CoM-Mixing training strategy (Appendix B).
>
> 3.  **More ablation and analyses**, such as data quality and data scaling experiments, detailed performance breakdowns of the Core-Cocktail training strategy, and investigations into the grouping factor's impact on GPU hours (Appendix C).
>
> 4.  Discussions of the limitations of the work and future plans (Appendix D).
>
>
> This deliberate organization allows us to present focused content clearly in the main text while providing supplementary details in the appendices, all in adherence to the conference page limits.

---

> > ### Author Response · Authors · 2025-11-27
> > **Response to Reviewer mDyG [2/3]**
> >
> > > **Q3:** Computational savings in terms of FLOPs of RTF by grouping of audio tokens
> >
> > Thank you for the valuable suggestion. The original submission provides quantitative analysis of the computational savings from grouping of audio tokens:
> >
> > 1.  **Training Efficiency:** Section 4.3 Line 453-454 and Figure 2 demonstrate a **~50% reduction in GPU training time from grouping than no grouping**, which correlates directly to the computational load.
> >
> > 2.  **Frame Rate:** Table 2 reports an LLM frame rate of **5Hz** in DrVoice (vs. 12.5Hz or 25Hz from the compared baselines). Since the attention complexity is quadratic with respect to the input sequence length, a **2.5x-5x reduction in the token sequence length implies a substantial reduction in FLOPs** **per second of the audio generated**.
> >
> >
> > We also computed the **real-time factor (RTF) per GPU during training**:
> >
> > *   When the group size is 1 (no grouping), the RTF is 0.197
> >
> > *   When the group size is 5, the RTF is substantially decreased to **0.089**, a **54.8%** relative reduction in RTF, which also corresponds to the **~50% reduction in GPU training time.**
> >
> >
> > > **Q4: Explaination of very large improvement from DrVoice to the 2nd best Kimi-Audio on Big Bench Audio.**
> >
> > Thank you for the valuable question.
> >
> > In the original submission, as shown in Table 2 main results, on Big Bench Audio (BBA), DrVoice substantially outperforms the second-best Kimi-Audio by **+11.1 (20.1% relative)** on Overall performance. This remarkable performance of DrVoice on BBA is a direct result of (1) our high quality and diverse instruction data and (2) our multi-stage Core-Cocktail Training strategy, which systematically enhances the model's reasoning and complex understanding capabilities—the very skills BBA is designed to evaluate.
> >
> > Our step-by-step performance tracking and analysis clearly validates the above explanation (the definitions of the two stages in Core-Cocktail Training are provided in Section 3.3 Core-Cocktail Training in the original submission):
> >
> > 1.  Stage 1: After this initial fine-tuning stage using our high-quality data mixture, our model achieves the Overall score of **57.2** on BBA. This performance is already a **solid 2.00-point lead** over the 2nd-best Kimi-Audio (55.2), establishing a strong foundation for the following training. We attribute this strong performance of DrVoice after Stage 1 training to the high quality and diversity of our instruction data (Appendix C, Data Quality). Note that with fixing the initialization mismatch for SRH as explained in our "Global Response to All Reviewers", DrVoice's performance on BBA after Stage 1 is improved to **58.75**.
> >
> > 2.  Stage 2: After Stage 2, the performance on BBA is boosted from **57.2** to **66.25 in the original submission.** The Core-Cocktail Training strategy is highly effective because it re-integrates the robust knowledge and reasoning abilities of the base LLM, mitigating catastrophic forgetting and enabling more stable, precise optimization. Stage 2 in Core-Cocktail Training is crucial for enhancing the model's core reasoning capabilities, which directly translates to a substantial performance gain on BBA's complex reasoning tasks. Note that with fixing the initialization mismatch for SRH as explained in our "Global Response to All Reviewers", DrVoice's performance on BBA after Stage 2 is improved to **68.7**.
> >
> > 3.  Stage 3 (Reinforcement Learning): As described in our "Global Response to All Reviewers", after submission, to ensure fairer comparisons with competitive baselines that applied RL, we also employ RL, including DPO and GSPO, to specifically sharpen the model's instruction-following and reasoning abilities on complex user queries. This targeted training further refines the model's capabilities, elevating the Overall score on BBA from **68.7** to **74.05**.
> >
> >
> > In summary, **in the original submission**, the 20% relative gain on Overall score from DrVoice to Kimi-Audio on BBA (**66.3** vs. 55.2) can be attributed to the synergistic combination of (1) our high quality and diverse instruction data and (2) our multi-stage Core-Cocktail Training strategy, where each stage progressively builds and refines the advanced reasoning skills that this particular benchmark particularly evaluates. _After submission_, fixing the initialization mismatch for SRH and adding RL further boosts DrVoice's performance to **74.05**.

---

> > > ### Author Response · Authors · 2025-11-27
> > > **Response to Reviewer mDyG [3/3]**
> > >
> > > > **Q5: Ethics concerns regarding Kimi-Audio.**
> > >
> > > 1.  We would like to emphasize that **DrVoice is a completely independent project and not a follow-up to Kimi-Audio.** We did not use their checkpoints, code, or any internal data during the development of our model. Our use of the Kimi-Audio checkpoints, the same as for all other baselines, was strictly for the purpose of rigorous comparative evaluations on public benchmarks—a common and necessary practice in academic research.
> > >
> > > 2.  We would like to emphasize that **DrVoice is fundamentally different from Kimi-Audio and innovative in its own right**, both in architecture and training methodology.
> > >
> > >
> > > *   **Model Architecture:**  The core of DrVoice is our novel **Dual-Resolution Speech Representations (DRSR)**. While Kimi-Audio incurs significant computational costs by operating at a 12.5Hz frame rate, DrVoice introduces a **grouping mechanism** to reduce the LLM's input frame rate to a highly efficient **5Hz**, paired with a dedicated **Speech Refined Head (SRH)** for generating high-quality speech.
> > >
> > > *   **Training Methodology:** We introduce two novel training strategies: **Core-Cocktail Training**, which effectively preserves the base LLM's knowledge, including reasoning capabilities, during fine-tuning, and **CoM-Mixing Training**, which provides a structured curriculum to enhance modality alignment and complex reasoning.
> > >
> > >
> > > These combined innovations are distinct from Kimi-Audio and are central to our model's strong performance.
> > >
> > > 1.  We respectfully disagree on the comment that "the evaluations in Table 2, the numbers are very close to each other in most cases". Based on our original submission, the Reviewer already noticed and commented on the substantial, 20% relative improvement from DrVoice to Kimi-Audio on Big Bench Audio; also, **on UltraEval-Audio (S2S), DrVoice also substantially outperforms Kimi-Audio (47.66 over 42.79). These substantial improvements are far from "very close to each other in most cases".**
> > >
> > >
> > >       Moreover, as elaborated in our "Global Response to All Reviewers", after fixing the initialization mismatch of SRH and applying RL,  DrVoice's updated results in Table 2  further demonstrate that **DrVoice substantially outperforms Kimi-Audio and all other baselines, establishing a new SOTA on all four benchmarks**. Specifically:
> > >
> > > *   On **OpenAudioBench**, DrVoice is ahead the second-best (Kimi-Audio) by **3.0 points**.
> > >
> > > *   On **VoiceBench**, DrVoice leads the runner-up (Kimi-Audio) by **3.2 points**.
> > >
> > > *   On **UltraEval-Audio**, DrVoice leads the runner-up by **6.2 points** and Kimi-Audio by **13.8 points**.
> > >
> > > *   On **Big Bench Audio**, DrVoice achieves a remarkable **18.2-point** lead over the runner-up (MiniCPM-o 2.6) and **18.8-point lead** over Kimi-Audio.
> > >
> > >
> > > **This consistent and significant performance advantage across all four benchmarks, driven by our unique architecture and training strategies, clearly distinguishes DrVoice as a novel and superior contribution to the field**.

---

> > > ### Comment · Reviewer_mDyG · 2025-11-27
> > >
> > > For Q4, Thanks for explaining the reasons behind the good performance on the BBA task. However, I am not concerned about that, instead, I was a looking for a clearer explanation of the BBA task itself and what is different in this task as compared to other benchmarks mentioned in the text. The response still does not address this question.

---

> > > > ### Author Response · Authors · 2025-11-28
> > > >
> > > > Thank you for the follow-up question and for giving us the opportunity to clarify. The primary difference is that **Big Bench Audio (BBA) is specifically designed to test a model's complex, multi-step reasoning abilities through an audio medium**, whereas the other benchmarks primarily evaluate broader audio-language capabilities such as comprehension, instruction following, and conversational quality.
> > > >
> > > > As described by its creators, BBA is an audio version of a subset of the challenging Big Bench Hard questions, explicitly targeting the reasoning capabilities of audio-processing models. It is composed of four distinct and difficult reasoning tasks:
> > > >
> > > > 1.  **Formal Fallacies Syllogisms Negation:** Tests the ability to perform rigorous logical deduction from a given context.
> > > > 2.  **Navigate:** Evaluates spatial and sequential reasoning by requiring the model to track an agent's position through a series of steps.
> > > > 3.  **Object Counting:** Requires semantic categorization and arithmetic, such as counting the total number of "fruits" when given a list of individual items (e.g., "three pianos, two strawberries, one table"). This is not simple counting but requires knowledge of object hierarchies.
> > > > 4.  **Web of Lies:** Assesses the model's capacity to evaluate the truth value of a complex Boolean function expressed as a natural-language word problem.
> > > >
> > > > In contrast, the other benchmarks focus on different skills:
> > > > *   OpenAudioBench and VoiceBench are broader in scope, evaluating aspects like open-ended question-answering (e.g., TriviaQA), conversational ability (e.g., AlpacaEval), and knowledge retrieval based on spoken queries. While they contain reasoning components, they are not as formally structured or as focused on multi-step abstract logic as BBA.
> > > > *   UltraEval-Audio primarily assesses the end-to-end quality of speech-to-speech interactions, evaluating both the correctness of the response and the quality of the generated speech.
> > > >
> > > > In short, BBA's tasks are designed such that the answer cannot be directly found or easily inferred from the prompt's surface text. It requires a deeper, more abstract chain of reasoning. This focus on **rigorous, formal reasoning** is why DrVoice's strong performance on BBA is particularly noteworthy.

---

> > > ### Comment · Reviewer_mDyG · 2025-11-27
> > >
> > > For Q3, thanks for sharing these additional statistics on the computational savings.

---

### Official Review · Reviewer_EyEh · 2025-10-31

**Soundness:** 3
**Presentation:** 3
**Contribution:** 2
**Rating:** 4
**Confidence:** 4

**Summary:**

The author proposed a 7B speech text voice conversion model, DrVoice, that leverages joint autoregressive modeling and dual speech representations.

The paper contributed on both model architectures and the training strategies. The model introduces a mechanism to alleviates the temporal resolution mismatch between text and speech tokens and reduce the computational cost. Specifically, it applied several grouping/ungrouping techniques to map 25 Hz audio tokens to 5Hz speech tokens and combined it with text tokens. Also, it uses a speech refined head(SRH) to preserve the intermediate semantic and acoustic representation during token generation. On the training side, two training strategies are applied: chain-of-modality(CoM) mix training for generating structured thoughts being used as language model text prompt, and a two-stage training method called Core-Cocktail Training for refining model's performance during training with large learning rate.

The model was trained on 100k hours of audio-tex paried data and evaluated with comprehensive benchmark metrics comparing to multiple baselines including previous SOTA. The author claimed their model achieved state of the art on OpenAudioBench and Big Bench Audio benchmarks, and performance comparable to SOTA on VoiceBench and UltraEval-audio benchmarks.

**Strengths:**

The model extended existing parallel joint speech-text model by the new methods below:
1. The author contributed an idea of reducing the temporal resolution of extracted audio representations from 25Hz to 5Hz to match the 3Hz text representation using speech token grouping. This benefits to reduce the computational costs and alleviate the frequency discrepancy between different representations. Leveraging the fine-grained acoustic information during generative scenarios is intuitive. The evaluation shows the speech quality is not compromised.
2. The author proposed to use both CoM-mixing and Core-Cocktail Training for model performance during training phase. The chain-of-modality mixing training allows the model to summarize intermediate textual transcriptions before feeding into LLM. The Core-Cocktail Training contributes to model training with large learning rate while preserving performance.
3. The SRH is proved useful for speech generation in the ablation study.

**Weaknesses:**

1. Although the grouping mechanism helps with reducing the temporal resolution and alleviate two representations, the ungrouping and the Speech Refined Head is making the model architecture more complicated and may worsen the model's speed due to its auto-regressive nature.
2. In the benchmark results, the model doesn't beat all the baselines in OpenAudioBench. The author claimed that the average score is the highest, but it is also true that it is not showing comparable perforamance on some tasks such as Reasoning QA(57.92 vs 63.76 Qwen2.5). There are not enough evidence showing that DrVoice is a new SOTA on openAudioBench.
3. Similarly, for VoiceBench and UltraEval-Audio, DrVoice only shows best perforamance on 1 or 2 tasks. While Kimi-Audio as a former SOTA still dominates on 5 out of 8 tasks. It could be better to discuss the performance comparison more.

**Questions:**

1. Does the ungrouping(linear + split) and SRH cause any increased computational cost? How does it compare to the effect of reducing temporal resolution?
2. From the ablation study, the ungrouping and SRH contributes more to S2M and T2M, while it doesn't help much with T2T. Is SRH making redundancy for T2T tasks?

---

> ### Author Response · Authors · 2025-11-27
> **Response to Reviewer EyEh**
>
> We sincerely appreciate your thoughtful reviews and valuable feedback. Please first check our "**Global Response to All Reviewers**".  We hope the following point-by-point response could fully address all your remaining concerns and questions.
>
> > **Q1:** the ungrouping and the Speech Refined Head may worsen the model's speed**.**
>
> We agree that SRH adds a component to the architecture, but it is a deliberate design choice to solve the **quality-efficiency trade-off**. Simple grouping (reducing resolution) allows the massive LLM backbone to run efficiently but loses acoustic details. The SRH acts as a lightweight, specialized module to restore the acoustic details. The computational overhead of the SRH is **marginal** compared to the substantial computational savings in the LLM backbone from grouping: by reducing the sequence length by 5x from the LLM frame rate of **5Hz** in DrVoice compared to the common 25Hz in baselines, the quadratic attention cost in the 7B-parameter backbone drops dramatically. **In the original submission, as presented in Line 453-454 in the main text, and also shown in Appendix C Figure 2, using a grouping factor of 5 together with ungrouping and SRH reduces total GPU training hours by nearly 50% compared to no grouping.**
>
> > **Q2: SOTA claims on OpenAudioBench, VoiceBench, and UltraEval.**
>
> Please refer to our "**Global** **Response to All Reviewers**" and the updated Table 2 in the revised manuscript. With our fix in SRH initialization and RL integration,
>
> *   **OpenAudioBench:** DrVoice is now **SOTA on all tasks** except on ReasoningQA (59.90 vs Qwen2.5-omni's 63.76)**,**  with Overall score surpassing the second best Kimi-Audio by **3.0 points**.
>
> *   **UltraEval Audio:** DrVoice now achieves **SOTA on all tasks of UltraEval-Audio**, with **Overall score +6.2 over the runner-up Qwen2.5-omni**.
>
> *   **Voice Bench**: DrVoice is now **SOTA on five out of 8 tasks of VoiceBench** and close second-best on the remaining 3 tasks, with **Overall score +3.2 points over the runner-up Kimi-Audio**.
>
> *   **Big Bench Audio**: DrVoice achieves **SOTA performance** and substantially outperforms the runner-up **on each task, with Overall score +18.2** over the second best MiniCPM-o 2.6.
>
>
> In summary, _with our fixing the initialization mismatch in SRH and integrating RL_, **DrVoice is SOTA on all four benchmarks and on most tasks,**  and its advantages over baselines like Kimi-Audio and Qwen2.5-omni have been significantly increased.
>
> > **Q3: Computational cost of Ungrouping/SRH vs. Grouping benefits.**
>
> Thank you for the question. The ungrouping process and the SRH indeed introduce a moderate computational overhead, as confirmed by our ablation study. Specifically, the inclusion of SRH increases the training time by 430 GPU hours (from 1090 to 1520 hours for a grouping factor of 5). However, this increase is **marginal** compared to the **substantial computational savings** achieved by reducing the temporal resolution via grouping. By increasing the grouping factor from 1 to 5, we reduce the total computational cost from 3360 GPU hours to 1520 GPU hours—a net reduction of 1840 GPU hours, or **54.8% relative reduction**. This result demonstrates that **our dual-resolution architecture strikes a highly effective quality-efficiency balance, where the immense efficiency gains from grouping far outweigh the minor cost of the SRH needed for high-fidelity speech synthesis**.
>
> > **Q4: Is SRH redundant for T2T tasks?**
>
> Thank you for the insightful question. You are correct that the SRH's primary and most dramatic impact is on speech-generation tasks (see the impact on S2M and T2M in Table 4). However, **SRH is also a critical and non-redundant component for T2T tasks.**The ablation in Table 4 in the original submission shows removing SRH (w/o. SRH) degrades T2T performance from 74.00 (w/o. CSE) to 73.00. While slight, this non-negligible impact indicates that SRH is an integral part of the jointly trained system. More importantly, this result verifies a key strength of our design: **SRH successfully decouples high-fidelity speech synthesis from the core text generation pathway**, which **ensures that adding powerful speech capabilities does not compromise the model's text reasoning abilities**, a common and significant challenge in joint speech-text models.

---

### Official Review · Reviewer_6YdV · 2025-10-31

**Soundness:** 3
**Presentation:** 3
**Contribution:** 2
**Rating:** 6
**Confidence:** 4

**Summary:**

This paper introduces DrVoice, a parallel speech-text voice conversation model built upon a large language model (LLM). The core contribution is a novel Dual-Resolution Speech Representation (DRSR) mechanism designed to address the temporal resolution mismatch between speech and text tokens in end-to-end (E2E) spoken dialogue systems. Specifically, the model groups higher-frequency (25Hz) input speech tokens into lower-frequency (5Hz) representations before feeding them to the LLM, reducing computational cost and aligning the speech processing rate more closely with that of text. The authors also propose two training strategies: a CoM-Mixing strategy to train the model on various interaction patterns (e.g., text-only, speech+text), and a Core-Cocktail training strategy to mitigate catastrophic forgetting of the base LLM's capabilities. Experimental results show that DrVoice-7B achieves good performance on OpenAudioBench and Big Bench Audio, and is competitive on other benchmarks like VoiceBench and UltraEval-Audio, establishing it as a strong open-source model in its size class.

**Strengths:**

1. The central idea of using dual-resolution speech representations is well-motivated and addresses a key, practical problem in joint speech-text modeling: the significant discrepancy in token rates between speech and text. The proposed grouping/ungrouping mechanism is an elegant solution that directly tackles this issue, leading to significant computational efficiency gains (processing at 5Hz within the LLM) without sacrificing output quality, thanks to the Speech Refined Head.

2. The paper presents a comprehensive set of experiments on multiple challenging benchmarks. DrVoice achieves SOTA results on two benchmarks (OpenAudioBench, Big Bench Audio) and demonstrates performance on par with other leading models on VoiceBench and UltraEval-Audio. This strong and balanced performance across tasks involving both understanding and generation solidifies the effectiveness of the proposed architecture.

3. The reduction of the LLM's operating frame rate from the common 12.5Hz or 25Hz to 5Hz is a major practical advantage. As shown in the ablation studies (Figure 2 in appendix), this leads to a nearly 50% reduction in GPU hours for training. This makes training and deploying such powerful spoken language models more feasible and is a valuable contribution to the field.

4. The proposed training strategies, CoM-Mixing and Core-Cocktail, are thoughtful additions. CoM-Mixing provides a structured curriculum for learning complex, multi-stage reasoning (think-then-speak), while the Core-Cocktail method offers a pragmatic approach to balance new learning with knowledge retention from the base LLM. The ablation studies effectively demonstrate the positive impact of these techniques.

**Weaknesses:**

1. The paper notes that DrVoice's ASR-WER (11.2) is higher than that of Qwen2.5-Omni (3.48), suggesting weaker text-speech alignment. The authors hypothesize this is because Qwen2.5-Omni feeds text directly into its "Talker" module, while DrVoice only uses hidden states. This is a crucial architectural trade-off. While the proposed solution (adding text as input to SRH) is mentioned as future work, the current limitation is significant. High ASR-WER can indicate issues with intelligibility or word-level alignment, which is a critical aspect of a voice conversation model.

2. A large portion of the training data (3B assistant tokens) is synthesized using CosyVoice. While the paper uses WER filtering, training on synthetic data can introduce biases from the teacher model and may not fully capture the diversity and nuances of real human speech. The paper could benefit from a discussion on the potential limitations of this approach and how it might affect the model's performance on in-the-wild, real-world conversations.

3. While the dual-resolution mechanism is a clever and effective engineering contribution, the overall architectural design of DrVoice bears a strong resemblance to existing models, particularly the Qwen-Omni series. The core "Thinker-Talker" paradigm, where a central LLM ("Thinker") generates intermediate representations that are then passed to a separate speech synthesis module ("Talker"), is conceptually very similar to DrVoice's structure of a shared LLM layer feeding into a dedicated Speech Refined Head (SRH).

**Questions:**

1. Could you clarify the training process for the Speech Refined Head (SRH)? Is the SRH jointly pre-trained with the LLM backbone, allowing gradients from the SRH loss to propagate back to the LLM? Or is the SRH trained separately, merely using the frozen LLM's hidden states as input? Have you explored both strategies, and what is their comparative impact on performance?

2. In the ablation study presented in Table 4, you analyze the impact of removing the Continuous Speech Encoder (CSE) and the Speech Refined Head (SRH). Could you please specify the alternative mechanisms used in these ablated models? For the "w/o. CSE" setting, how are the raw speech inputs processed and fed into the LLM? For the "w/o. SRH" setting, how are the speech output tokens generated from the shared LLM layer's hidden states? Is a simpler projection layer used instead?

3. Table 3 shows that DrVoice has a relatively high ASR-WER of 11.2, which is significantly worse than top-performing models like Qwen2.5-Omni (3.48). This suggests a potential weakness in speech-text alignment or intelligibility. How do you interpret this result, and what specific architectural or training factors do you believe contribute to this performance gap?

---

> ### Author Response · Authors · 2025-11-27
> **Response to Reviewer 6YdV [1/2]**
>
> We sincerely appreciate the Reviewer's thoughtful review and constructive feedback. Please first check our "**Global Response to All Reviewers**". We hope our response below could address all of your remaining questions and concerns.
>
> > **Q1 / Q6: ASR-WER (11.2) vs. Qwen2.5-Omni (3.48) and architectural trade-offs.**
>
> Thank you for highlighting this critical comparison. We have addressed this in the revision:
>
> 1.  **Improved Alignment:** As explained in our "Global Response to All Reviewers", by fixing the SRH initialization mismatch (aligning pre-training tasks with the downstream objectives), we reduced the ASR-WER from 11.2 to **8.36** (see the updated Table 3). This ranks DrVoice second among baselines.
>
> 2.  **Architectural Context:** As explained in Line 425-431 in the paper, we acknowledge the ASR-WER gap between DrVoice and Qwen2.5-Omni, which stems from a fundamental paradigm difference. While Text-Driven models like Qwen2.5-Omni excel at text-to-speech fidelity due to their _one-way 'Thinker-to-Talker' flow_ (feeding text directly to its “Talker” module), this one-way flow makes their core LLM 'deaf' during generation. In contrast, DrVoice, as a Parallel Joint Model, creates a closed feedback loop by feeding speech tokens back into the LLM. **This architectural choice**, while presenting more challenge to alignment, **is a prerequisite for true full-duplex voice interaction**.
>
> 3.  **Semantic Performance:** Despite the ASR-WER difference, DrVoice substantially outperforms Qwen2.5-Omni on speech generation tasks, as shown by DrVoice's much better performance on each task in the UltraEval-Audio Benchmark (S2S) (**Overall score 56.66 over Qwen2.5-omni's 50.46**) and the Big Bench Audio (S2S) (**71.2 on S2S over Qwen2.5-omni's** **53.6**).
>
> 4.  **Quality vs. Alignment:** While ASR-WER tracks text match, **UTMOS** tracks audio quality. Our UTMOS score (**4.29**) is in fact slightly higher than Qwen2.5-Omni (4.28), indicating **high-fidelity speech generation despite the alignment challenges**.
>
> 5.  **Future Mitigation:** DrVoice tends to generate more complex and structurally varied text than baselines, which inherently increases alignment difficulty compared to simple TTS transcripts. To further close the WER gap, currently we are (1) scaling up the SRH and (2) implementing text-rewriting strategies to better align the target text with speech.
>
>
> > **Q2: Reliance on synthetic data (CosyVoice) and potential bias/limitations.**
>
> We agree that synthetic data alone is insufficient for real-world robustness. In the original submission, we have implemented a three-pronged approach to mitigate this bias, as emphasized and clarified in the revised manuscript:
>
> 1.  **Robust Encoder:** As explained in Line 269, we utilize the **Whisper Encoder**, which is pre-trained on vast amounts of real-world noisy audio, providing a robust foundation for speech understanding.
>
> 2.  **Real Data Mixing**: In Stage 2 training of the Core-Cocktail training strategy (Line 917-941 and Table 6), we augment the training data with real-world ASR data (Stage 2 w/. ASR data in Table 6). This strategy increased our performance on CommonEval (a benchmark based on **human speech**) from 3.35 to **3.77**. We attribute this gain to the fact that ASR data exposes the model to the natural disfluencies, varied intonations, and ambient noise present in real-world speech, enhancing its robustness.
>
> 3.  **Reinforcement Learning (RL):** To further bridge the gap between synthetic data and real-world data, we selected real-world speech queries from open-source ASR datasets. We used our SFT model to generate T2T and S2T response. Using Direct Preference Optimization (DPO) with these pairs—treating text-input generation as positive and speech-input generation as negative —we further improved the CommonEval score to **4.08**, a close second-best to MiniCPM-o 2.6's 4.15.

---

> > ### Author Response · Authors · 2025-11-27
> > **Response to Reviewer 6YdV [2/2]**
> >
> > > **Q3: Novelty compared to Qwen-Omni ("Thinker-Talker").**
> >
> > Thank you for the valuable question. While there are superficial similarities, **the architectures are fundamentally different**. Following the taxonomy in _Slam-Omni_ \[1\], DrVoice belongs to the **Parallel** paradigm, whereas Qwen-Omni adopts a **Text-Driven** paradigm. In Qwen-Omni, the "Talker" is conditioned on the _completed text output_ of the "Thinker".  In DrVoice, _generated speech tokens are fed back into the LLM autoregressively_; this allows the LLM to perceive the state of speech generation **in real-time**, enabling true cross-modal interaction and future applications such as **low-latency full-duplex communication**.
> >
> > > **Q4: Clarification on SRH training process (Joint vs. Separate).**
> >
> > Thank you for the question. We have clarified the SRH training process in Section 3.3 and Appendix C in the revised manuscript.
> >
> > 1.  **Pre-training:**  Please find our post-submission fix to pre-training SRH in "**Global Response to All Reviewers**" under "**Fixing the initialization mismatch in SRH**"
> >
> > 2.  **SFT Stage:** After pre-training SRH, the **entire model** (Speech Encoder, Adapter, Shared LLM, Text Head, and SRH) **is jointly fine-tuned**. Gradients propagate through the whole network.
> >
> > 3.  **Pre-training with LLM Frozen vs. Joint Pre-training of SRH and LLM:** We utilize pre-training SRH with the LLM frozen, since joint pre-training of SRH and LLM (1) may hinder SRH training due to its significantly smaller capacity compared to the LLM, and (2) may lead to knowledge forgetting in the LLM caused by the distribution of TTS data used for pre-training both SRH and LLM.
> >
> >
> > > **Q5: Details on ablation study (w/o CSE, w/o SRH).**
> >
> > Thank you for the suggestion. We have added these details to the experimental setup in the revised manuscript:
> >
> > *   **w/o CSE (Continuous Speech Encoder):** In this setting, user audio input is processed solely using the discrete speech tokenizer (same as the assistant's output tokenizer), without the Whisper features.
> >
> > *   **w/o SRH (Speech Refined Head):** To handle the resolution mismatch when SRH is removed, the hidden state from the shared LLM is passed through a projection layer ($d\_{model} \to k \times d\_{speech\\_token}$) and then split into $k$ tokens. This setup simulates a standard parallel prediction approach without autoregressive refinement.
> >
> >
> > ### References
> >
> > \[1\] Wenxi Chen, Ziyang Ma, Ruiqi Yan, Yuzhe Liang, Xiquan Li, Ruiyang Xu, Zhikang Niu, Yanqiao Zhu, Yifan Yang, Zhanxun Liu, Kai Yu, Yuxuan Hu, Jinyu Li, Yan Lu, Shujie Liu, Xie Chen, SLAM-Omni: Timbre-Controllable Voice Interaction System with Single-Stage Training. ACL (Findings) 2025: 2262-2282

---

### Author Response · Authors · 2025-11-27
**Global Response to All Reviewers [1/2]**

We sincerely thank all the reviewers for their careful reviews and constructive feedback. We appreciate the positive remarks from all three reviewers highlighting our contributions:

1.  **Reviewer 6YdV** appreciates the **dual-resolution speech representation (DRSR)** as a "well-motivated and elegant solution" to the temporal mismatch problem, noting its **significant computational efficiency gains** and reduction in GPU hours. The reviewer also praises the **thoughtful design of CoM-Mixing and Core-Cocktail training strategies**, as well as the strong experimental results demonstrating SOTA performance on OpenAudioBench and Big Bench Audio.

2.  **Reviewer EyEh** highlights the speech-token grouping mechanism's ability to reduce computational cost while maintaining speech quality. The reviewer also values the **intuitive fine-grained acoustic representation** and the effectiveness of **CoM-Mixing and Core-Cocktail Training**, while acknowledging the model's comprehensive evaluation across benchmarks.

3.  **Reviewer mDyG** describes the **use of both continuous and discrete audio representations** as a key strength and acknowledges the novelty of **CoM-Mixing and Core-Cocktail Training**, noting **competitive performance** and SOTA results on Big Bench Audio tasks. The reviewer also appreciates the computational cost reduction approach.


We are grateful for these encouraging comments validating the novelty, efficiency, and strong performance of our proposed DrVoice model.

A common concern is shared among reviewers, namely the consistency of the proposed model’s performance across different benchmarks, particularly where it falls short on some baselines (e.g., in tasks like Reasoning QA on OpenAudioBench or certain tasks in VoiceBench and UltraEval-Audio). Below we address this shared concern.


## Updated Benchmark Results and Enhanced Performance

After the original submission, we identified an initialization mismatch in the Speech Refined Head (SRH) and rectified the problem to ensure that pretraining of SRH closely matches the downstream task format. This correction substantially improves DrVoice's performance on Speech-to-Speech (S2S) tasks.

Moreover, it is important to note that both Qwen2.5-Omni and Kimi-Audio baselines employ reinforcement learning (RL) to enhance capabilities of their models; whereas, in our original submission, DrVoice only applied SFT and did not employ any RL. Therefore, we also add an RL stage after SFT to enable a fairer comparison with the competitive baselines and provide a fairer evaluation of DrVoice's potentials.

1.  **Fix the initialization mismatch in SRH**

    In the original submission, we independently trained a Text-to-Speech (TTS) task using the Qwen2.5-0.5B model in a Text-to-Multimodal format (Text -> joint speech-text response) with a 25Hz token rate. The resulting trained weights were subsequently loaded onto the SRH. However, the initialization strategy exhibited some mismatches with the subsequent Stage 1 and Stage 2 training processes, as the text input was first processed through the shared LLM layer before reaching the SRH.

    To address the initialization mismatch in the SRH, we trained it exclusively with a streaming TTS objective, wherein the model predicts speech token based on a streaming feed of joint speech-text inputs, rather than the full text sequence while freezing Speech Encoder and LLM.  This way, the SRH is trained to closely match the downstream task format.  We observe a **substantial performance improvement in S2S tasks**, as UltraEval-Audio S2S scores are improved from 47.66 to **57.20****.**

2.  **Add an RL stage after SFT**


We leverage Reinforcement Learning (RL) to enhance robustness in real-world speech comprehension while improving instruction-following and reasoning capabilities.

*   **Real-world Speech Comprehension:** To address the scarcity of real speech data, a preference dataset was constructed using open-source ASR datasets. A text LLM was employed to identify samples suitable for user queries, with canonical responses generated by another text LLM. The DrVoice-SFT model processed the original speech queries to establish the preference data. We utilize Direct Preference Optimization (DPO) to enhance real-world speech comprehension abilities.

*   **Instruction-Following:** To improve instruction-following capabilities, an open-source preference dataset tailored for instruction-following tasks is utilized. DPO is applied to strengthen the model's ability to accurately follow instructions.

*   **Reasoning:** For enhanced reasoning capabilities, GSPO \[1\] training is applied to mathematical problems, enabling the model to excel in problem-solving tasks.


These enhancements led to substantial improvements across key benchmarks and equipped the model with stronger generalization and task-specific performance.

---

> ### Author Response · Authors · 2025-11-27
> **Global Response to All Reviewers [2/2]**
>
> The **revised paper** has been updated these new results.
>
> | Benchmark | Previous SOTA | **DrVoice(Original Submission)** | **DrVoice+fix-SRH-mistach** | **DrVoice++RL** |
> | --- | --- | --- | --- | --- |
> | _OpenAudioBench (S2T)_ |  |  |  |  |
> | AlpacaEval | 77.90 | 78.34 | 81.21 | 82.61 |
> | Llama Q. | 79.33 | 80.33 | 84.67 | 83.00 |
> | Reasoning QA | 63.76 | 57.92 | 53.47 | 59.90 |
> | TriviaQA | 63.00 | 61.50 | 63.40 | 64.50 |
> | Web Q. | 70.20 | 68.10 | 70.20 | 70.20 |
> | Overall | 69.08 | 69.24 | 70.59 | 72.04 |
> | _VoiceBench (S2T)_ |  |  |  |  |
> | AlpacaEval | 4.50 | 4.52 | 4.57 | 4.74 |
> | CommonEval | 4.15 | 3.77 | 3.87 | 4.08 |
> | SD-QA | 63.12 | 68.54 | 63.29 | 64.30 |
> | MMSU | 62.17 | 60.31 | 64.74 | 67.27 |
> | OpenBookQA | 83.52 | 79.56 | 81.54 | 82.20 |
> | IFEval | 61.10 | 59.30 | 60.70 | 71.39 |
> | AdvBench | 100.00 | 98.65 | 99.04 | 99.62 |
> | Overall | 76.93 | 76.02 | 76.87 | 80.17 |
> | _UltraEval-Audio (S2S)_ |  |  |  |  |
> | AlpacaEval | 58.69 | 49.65 | 63.48 | 59.29 |
> | Llama Q. | 67.67 | 68.00 | 76.67 | 75.33 |
> | TriviaQA | 40.52 | 35.35 | 43.85 | 46.09 |
> | Web Q. | 40.00 | 37.65 | 44.78 | 45.92 |
> | Overall | 50.46 | 47.66 | 57.20 | 56.66 |
> | _Big Bench Audio (S2T & S2S)_ |  |  |  |  |
> | S2T | 59.40 | 71.60 | 74.40 | 76.90 |
> | S2S | 55.40 | 60.90 | 63.00 | 71.20 |
> | Overall | 55.8 | 66.25 | 68.70 | 74.05 |
>
>
> **State-of-the-Art Performance across all 4 major benchmarks:** DrVoice now achieves SOTA results across all 4 major benchmarks as shown in Table 2 in the updated manuscript. Specifically,
>
> *   **OpenAudioBench:** DrVoice is now **SOTA on all tasks** except on ReasoningQA (59.90 vs Qwen2.5-omni's 63.76)**,**  with Overall score surpassing the second best Kimi-Audio by **3.0 points**.
>
> *   **Voice Bench**: DrVoice is now **SOTA on five out of 8 tasks of VoiceBench** and close second-best on the remaining 3 tasks, with **Overall score +3.2 points over the runner-up Kimi-Audio**.
>
> *   **UltraEval Audio:** DrVoice now achieves **SOTA on all tasks of UltraEval-Audio**, with **Overall score +6.2 over the runner-up Qwen2.5-omni**.
>
> *   **Big Bench Audio**: DrVoice achieves **SOTA performance** and substantially outperforms the runner-up **on each task, with Overall score +18.2** over the second best MiniCPM-o 2.6.
>
>
> We plan to open-source the related models and code after the paper is accepted.
>
> \[1\] Chujie Zheng, Shixuan Liu, Mingze Li, Xiong-Hui Chen, Bowen Yu, Chang Gao, Kai Dang, Yuqiong Liu, Rui Men, An Yang, Jingren Zhou, and Junyang Lin. Group sequence policy optimization. CoRR, abs/2507.18071, 2025. doi: 10.48550/ARXIV.2507.18071.
>
> ## The List of Revisions Made to the Manuscript
>
> To fully address all the questions and concerns from all the reviewers, we made the following major revisions in the updated manuscript (**all updated text is highlighted in cyan**):
>
> 1.  **Updated Benchmark Results Tables:** Tables 2 and 3 now reflect DrVoice's new SOTA results across all four benchmarks on most tasks, after fixing SRH's initialization mismatch and applying RL.
>
> 2.  **New Section on RL:** Added details on the implementation of DPO and GSPO (Group Sequence Policy Optimization) in Appendix C Reinforcement learning.
>
> 3.  **Clarified Pre-training:** Added Appendix C SRH-Pretraining Policy to clarify the SRH pre-training strategy.
>
> 4.  **Expanded Ablation:** Added definitions of "w/o CSE" and "w/o SRH" in Section 4.3.
>
> 5.  **Efficiency Metrics:** Emphasized the relationship between 5Hz frame rate and computational savings in Introduction and Conclusion.
>
>
> We hope that this revised manuscript addresses all of the questions and concerns raised by the reviewers; if not, we would be happy to engage in further discussions and revisions of the paper to best present our work.

---

### Author Response · Authors · 2025-12-03
**Summary for the Area Chair**

Dear AC,

We sincerely thank all reviewers for their insightful feedback, which has helped us significantly strengthen our manuscript. We think our rebuttal addressed all the reviewers' concerns.

**Summary of Rebuttal and Discussions**

A shared concern was the consistency of our model's performance. We addressed this directly in our **Global Response**. The updated results are due to two key post-submission updates: **(1) fixing a crucial initialization mismatch** in our Speech Refined Head (SRH) and **(2) adding a Reinforcement Learning (RL) stage** to ensure **fairer comparisons** with competitive baselines that also use RL. These updates led to DrVoice achieving new **State-of-the-Art (SOTA) performance across all four major benchmarks** (OpenAudioBench, VoiceBench, UltraEval-Audio, and Big Bench Audio), resolving the reviewers' initial concerns about performance gaps (see **Global Response** for detailed results).

We also addressed specific concerns:

*   We clarified that DrVoice’s superior performance on Big Bench Audio (BBA), which Reviewer mDyG noted was a "very large improvement" in the original submission, stems from our high-quality data and advanced Core-Cocktail training strategy designed for complex reasoning. The reviewer followed up to state he was **"not concerned about"** this strong performance, which validates our original claim that DrVoice demonstrated a substantial advantage over baselines like Kimi-Audio, far from being "very close to each other." We further clarified the unique, complex reasoning focus of the BBA benchmark (see **Response to R-mDyG, Q4 & follow-up**).

*   Regarding the ethics concern raised by Reviewer mDyG about our work being a follow-up to Kimi-Audio, we have clarified that **DrVoice is a completely independent project** with fundamentally different architectural and training innovations. Our new SOTA results, which show an even larger performance gap over Kimi-Audio, further underscore this distinction (see **Response to R-mDyG, Q5**).

*   We addressed concerns about ASR-WER and synthetic data by showing the SRH fix improved alignment and by detailing our use of real-world data mixing and RL to enhance robustness (see **Response to R-6YdV, Q1 & Q2**).

*   We clarified that the computational overhead of our SRH is minimal compared to the significant efficiency gains from our dual-resolution design, which reduces training GPU hours by nearly 50% (see **Response to R-EyEh, Q1 & Q3**).


**Summary of Innovations and Contributions**

The reviewers widely recognized our paper's core contributions:

1.  **Dual-Resolution Speech Representation (DRSR):** This is our central innovation for achieving high efficiency without sacrificing quality. Reviewer 6YdV praised it as a **"well-motivated and elegant solution"** that yields **"significant computational efficiency gains."** Reviewer EyEh highlighted that it **"reduce\[s\] computational costs"** while ensuring speech quality is **"not compromised."**

2.  **Novel Training Strategies:** Our **CoM-Mixing** and **Core-Cocktail Training** strategies were recognized as key to the model's strong performance. Reviewer 6YdV described them as **"thoughtful additions,"** and Reviewer mDyG acknowledged their novelty.


These innovations, validated by strong empirical results and positive reviewer feedback, establish DrVoice as a leading open-source speech foundation model in ~7B models that advances the state of the art in both performance and computational efficiency.

---

### Meta-Review · Area_Chair_12ho · 2026-01-03

**Summary:**

Concerns regarding benchmark performance and quantified computational savings have been largely addressed. The authors demonstrated notable efficiency gains from the dual-resolution representation. Benchmark numbers have been significantly improved in the rebuttal due to a rectified mismatch issue in the SRH training. Remaining concerns seem to be mostly about the novelty of the proposed approach, which are subjective in nature and difficult to address fully.

**Reviewer Concerns:**

Reviewer concerns were largely addressed with the exception of novelty concerns -- which I believe are valid but perhaps minor.

**Reviewer Scores:**

I believe the reviewers would have raised their ratings to right above borderline or even a weak accept, given the updated SOTA results. I agree with reviewer mDyG that novelty is limited but sufficient, but for this reason, I don't think ratings would have been much higher than a weak accept in any case.

---

### Decision · Program_Chairs · 2026-01-26

Accept (Poster)